# GATE: How to Keep Out Intrusive Neighbors

## Abstract

Graph Attention Networks (GATs) are designed to provide flexible neighborhood aggregation that assigns weights to neighbors according to their importance. In practice, however, GATs are often unable to switch off task-irrelevant neighborhood aggregation, as we show experimentally and analytically. To address this challenge, we propose GATE, a GAT extension that holds three major advantages: i) It alleviates over-smoothing by addressing its root cause of unnecessary neighborhood aggregation. ii) Similarly to perceptrons, it benefits from higher depth as it can still utilize additional layers for (non-)linear feature transformations in case of (nearly) switched-off neighborhood aggregation. iii) By down-weighting connections to unrelated neighbors, it often outperforms GATs on real-world heterophilic datasets. To further validate our claims, we construct a synthetic test bed to analyze a model's ability to utilize the appropriate amount of neighborhood aggregation, which could be of independent interest.

## 1 Introduction

Graph neural networks (GNNs) (Gori et al., 2005) are a standard class of models for machine learning on graph-structured data that utilize node feature and graph structure information jointly to achieve strong empirical performance, particularly on node classification tasks. Input graphs to GNNs stem from various domains of real-world systems such as social (Bian et al., 2020), commercial (Zhang & Chen, 2020), academic (Hamaguchi et al., 2017), economic (Monken et al., 2021), biochemical(Kearnes et al., 2016), physical (Shlomi et al., 2021), and transport (Wu et al., 2019) networks that are diverse in their node feature and graph structure properties.

The message-passing mechanism of GNNs (Kipf & Welling, 2017; Xu et al., 2019) involves two key steps: a transformation of the node features, and the aggregation of these transformed features from a node's neighborhood to update the node's representation during training. While this has proven to be largely successful in certain cases, it generally introduces some problems for learning with GNNs, the most notorious of which is over-smoothing (Li et al., 2018). The enforced use of structural information in addition to node features may be detrimental to learning the node classification task, as shown by recent results where state-of-the-art GNNs perform the same as or worse than multi-layer perceptrons (MLPs) (Gomes et al., 2022; Yan et al., 2022; Ma et al., 2022). This raises a pertinent question for the GNN research community: *How much neighborhood aggregation is needed?*. Naturally, the answer is: *It depends; on the input graph, the task at hand, and possibly any domain knowledge.* In fact, it is what we would ideally want a model to learn.

A popular standard GNN architecture that, in principle, tries to resolve this problem is the Graph Attention Network (GAT) (Veličković et al., 2018; Brody et al., 2022). By design, neighborhood aggregation in GATs is characterized by learnable coefficients that are intended to assign larger weights to more important neighboring nodes (including the node itself) in order to learn better node representations. However, the role of these parameterized coefficients in the learning process (or what they learn) is not fully understood. In the quest for this understanding, we conduct a simple experiment. Given informative features and irrelevant information in a node's neighborhood, GATs should ideally resort to assigning near-zero importance to neighbor nodes, effectively switching off neighborhood aggregation. However, we find that, counter-intuitively, GATs are unable to do this in practice and continue to aggregate the uninformative features in the neighborhood which impairs the performance of GAT, particularly with an increase in model depth.

We address the challenge faced by GAT to effectively determine how well a node is represented by its own features in comparison to the features of nodes in its neighborhood, i.e., distinguish between the relative importance of available node features and graph structure information for a given task.

Firstly, we provide an intuitive explanation for the problem based on a conservation law of GAT gradient flow dynamics derived by Mustafa & Burkholz (2022). Building on this insight, we present GATE, an extension of the GAT architecture that is able to switch neighborhood aggregation on and off as necessary. This allows our proposed architecture to gain the following advantages over GAT:

1. It alleviates the notorious over-smoothing problem by addressing the root cause of unnecessarily repeated neighborhood aggregation.

2. It allows the model to benefit from more meaningful representations obtained solely by deeper non-linear transformations, similarly to perceptrons, in layers where neighborhood aggregation is (nearly) switched off.

3. It often outperforms GATs on real-world heterophilic datasets by down-weighting connections to unrelated neighbors.

4. It offers interpretable learned self-attention coefficients, at the node level, that are indicative of the relative importance of feature and structure information in the locality of the node.

In order to validate these claims, we construct a synthetic test bed of two opposite types of learning problems for node classification where label-relevant information is completely present only in a node's i) own features and ii) neighboring nodes' features. GATE is able to adapt to both cases as necessary. On real-world datasets, GATE performs competitively on homophilic datasets and is substantially better than GAT on heterophilic datasets.

Our contributions are as follows:

- We identify and experimentally demonstrate a structural limitation of GAT, i.e., its inability to switch off neighborhood aggregation.

- We propose GATE, an extension of GAT, that overcomes this limitation and, in doing so, unlocks several benefits of the architecture.

- We update an existing conservation law relating the structure of gradients in GAT to GATE.

- We construct a synthetic test bed to validate our claims, which could be of independent interest given the active research along similar lines.

## 2 RELATED WORK

To relieve GNNs from the drawbacks of unnecessarily repeated neighborhood aggregation in deeper models, initial techniques were inspired by classical deep learning of MLPs such as normalization (Cai et al., 2021; Zhao & Akoglu, 2020; Zhou et al., 2020; 2021) and regularization (Papp et al., 2021; Rong et al., 2020; Yang et al., 202; Zou et al., 2019).

More recently, the need for deeper models and architectural changes to limit neighborhood aggregation as necessary has been recognized leading to approaches that use linear combinations of initial features and current layer representation (Gasteiger et al., 2019), add skip connections and identity mapping (Chen et al., 2020; Cong et al., 2021), combine representations of all previous layers at the last layers (Xu et al., 2018), aggregate information from a node-wise defined range of $k$-hop neighbors(Liu et al., 2020), and limit the number of aggregation iterations based on node influence scores (Zhang et al., 2021). However, these architectures are not flexible enough to benefit from utilizing additional network layers to simulate perceptron behavior, which, as we find, aids learning on heterophilic tasks. Ma et al. (2023) provide an insightful discussion on 'good' and 'bad' heterophily.

An orthogonal line of research uses graph structural learning (Yang et al., 2019; Stretcu et al., 2019; Franceschi et al., 2020) to amend the input graph structure such that neighborhood aggregation is beneficial for the given task. Such approaches are difficult to scale, more susceptible to over-smoothing, and potentially destroy any inherent information in the original graph structure. On the contrary, a standard GNN architecture empowered to selectively perform neighborhood aggregation avoids these pitfalls. Methods such as graph rewiring (Deac et al., 2022) to overcome other problems with GNNs such as over-squashing (Alon & Yahav, 2021) are complementary and may also

be combined with GATE. Additional supervision has also been proposed to improve the attention mechanism in GATs (Wang et al., 2019; Kim & Oh, 2021).

While we focus our insights on GAT, architectures based on GAT such as $\omega$GAT (Eliasof et al., 2023) also suffer from the same problem (see Fig. 10 in Appendix C). This further confirms that the universal problem with GAT has been correctly identified. In general, recent works direct effort to understand the current limitations of graph attention (Lee et al., 2023; Fountoulakis et al., 2023).

## 3 ARCHITECTURE

**Notation** Consider a graph $G = (V, E)$ with node set $\mathbb{V}$ and edge set $\mathbb{E} \subseteq \mathbb{V} \times \mathbb{V}$, where for a node $v \in \mathbb{V}$ the neighborhood is $\mathbb{N}(v) = \{u | (u, v) \in \mathbb{E}\}$ and input features are $\mathbf{h}_v^0$. A GNN layer updates each node's representation by aggregating over its neighbors' representation and combining it with its own features. The aggregation and combination steps can be performed together by introducing self-loops in $G$ such that, $\forall v \in \mathbb{V}, (v, v) \in \mathbb{E}$. We assume the presence of self-loops in $G$ unless specified otherwise. In GATs, this aggregation is weighted by parameterized attention coefficients $\alpha_{uv}$, which indicate the importance of node $u$ for $v$. A network is constructed by stacking $L$ layers, defined as follows, using a non-linear activation function $\phi$ that is homogeneous (i.e $\phi(x) = x\phi'(x)$) and consequently, $\phi(ax) = a\phi(x)$ for positive scalars $a$) such as ReLU $\phi(x) = \max\{x, 0\}$ or LeakyReLU $\phi(x) = \max\{x, 0\} + -\alpha \max\{-x, 0\}$.

**GAT** Given input representations $\mathbf{h}_v^{l-1}$ for $v \in V$, a GAT [1] layer $l \in [L]$ transforms those to:

$$\mathbf{h}_v^l = \phi \left( \sum_{u \in \mathbb{N}(v)} \alpha_{uv}^l \cdot \mathbf{W}_s^l \mathbf{h}_u^{l-1} \right), \quad \text{where} \tag{1}$$

$$\alpha_{uv}^l = \frac{\exp\left(e_{uv}^l\right)}{\sum_{u' \in \mathbb{N}(v)} \exp\left(e_{u'v}^l\right)}, \quad \text{and} \tag{2}$$

$$e_{uv}^l = \left(\mathbf{a}^l\right)^\top \cdot \phi\left(\mathbf{W}_s^l \mathbf{h}_u^{l-1} + \mathbf{W}_t^l \mathbf{h}_v^{l-1}\right) \tag{3}$$

The feature transformation weights $\mathbf{W}_s$ and $\mathbf{W}_t$ for source and target nodes, respectively, may also be shared such that $\mathbf{W}_s = \mathbf{W}_t$. We denote the weight-sharing variant of GAT by GAT$_S$.

**GATE** In addition, we propose GATE, a GAT variant that flexibly weights the importance of node features and neighborhood features. A GATE layer is also defined by Eq. (1) and (2) but modifies $e_{uv}$ in Eq. (3) to Eq. (4). Given that $q_{uv} = 1$ if $u = v$ and $q_{uv} = 0$ if $u \neq v$,

$$e_{uv}^l = \left((1 - q_{uv})\, \mathbf{a}_s^l + (q_{uv})\, \mathbf{a}_t^l\right)^\top \cdot \phi\left(\mathbf{U}^l \mathbf{h}_u^{l-1} + \mathbf{V}^l \mathbf{h}_v^{l-1}\right) \tag{4}$$

We denote $e_{uv}$ in Eq. (3) and (4) as $e_{vv}^l$ if $u = v$. For GATE, $\mathbf{W}_s^l$ in Eq. (1) is denoted as $\mathbf{W}^l$.

A weight-sharing variant of GATE, GATE$_S$, is characterized by all feature transformation parameters being shared in a layer (i.e. $\mathbf{W}^l = \mathbf{U}^l = \mathbf{V}^l$).

We next present theoretical insights into the reasoning behind the inability of GATs to switch off neighborhood aggregation, which is rooted in norm constraints imposed by the inherent conservation law for GATs. The gradients of GATE fulfill an updated conservation law (Thoerem 4.3) that enables switching off neighborhood aggregation in a parameter regime with well-trainable attention.

## 4 THEORETICAL INSIGHTS INTO NEIGHBORHOOD AGGREGATION

For simplicity, we limit our following discussion to GATs with weight sharing as they achieve a similar performance as GATs without weight sharing. Yet, similar arguments could also be derived for the latter case. The following conservation law was recently derived for GATs to explain trainability issues of standard initialization schemes. Even with improved initializations, we argue that this

---

[1] Throughout, we refer to GATv2 (Brody et al., 2022) as GAT for brevity.

law limits the effective expressiveness of GATs and hinders them from switching off neighborhood aggregation when necessary.

**Theorem 4.1** (Thm. 2.2 by Mustafa & Burkholz (2022)). *The feature weight and attention parameters $\mathbf{W}^l$ and $\mathbf{a}^l$ of a layer $l$ in a GAT network and their gradients fulfill:*

$$\langle \mathbf{W}^l[i,:], \nabla_{\mathbf{W}^l[i,:]}\mathcal{L} \rangle = \langle \mathbf{W}^{l+1}[:,i], \nabla_{\mathbf{W}^{l+1}[:,i]}\mathcal{L} \rangle + \langle \mathbf{a}^l[i], \nabla_{\mathbf{a}^l[i]}\mathcal{L} \rangle. \tag{5}$$

Intuitively, this equality limits the budget for the relative change of parameters and imposes indirectly a norm constraint on the parameters. Under gradient flow that assumes infinitesimally small learning rates, this law implies that the relationship $\left\|\mathbf{W}^l[i,:]\right\|^2 - \left\|\mathbf{a}^l[i]\right\|^2 - \left\|\mathbf{W}^{l+1}[:,i]\right\|^2 = c$ stays constant during training, where $c$ is defined by the initial norms. Other gradient-based optimizers fulfill this norm balance also approximately. Note that the norms $\left\|\mathbf{W}^l[i,:]\right\|$ generally do not assume arbitrary values but are determined by the required scale of the output. Deeper models are especially less flexible in varying these norms as deviations could lead to exploding or diminishing output and/or gradients. In consequence, the norms of the attention parameters are bounded also. Furthermore, a parameter becomes harder to change during training when its magnitude increases. This can be seen by transforming the law with respect to the relative change of a parameter defined as $\Delta\theta = \nabla_\theta \mathcal{L}/\theta$ for $\theta \neq 0$ or $\delta\theta = 0$ for $\theta = 0$ as follows.

$$\sum_{j=1}^{n_l}(\mathbf{W}_{ij}^{l-1})^2\Delta\mathbf{W}_{ij}^{l-1} = \sum_{k=1}^{n_l}(\mathbf{W}_{ki}^l)^2\Delta\mathbf{W}_{ki}^l + (\mathbf{a}_i^{l-1})^2\Delta\mathbf{a}_i^{l-1}. \tag{6}$$

The higher the magnitude of an attention parameter $(\mathbf{a}_i^l)^2$, the smaller will be the relative change $\Delta\mathbf{a}_i^l$ and vice versa, as explained by the next insight.

**Insight 4.2** (Effective expressiveness of GATs). GATs are challenged to switch off neighborhood aggregation during training, as this would require the model to enter a less trainable regime with large attention parameters $\|\mathbf{a}\|^2 >> 1$.

An intuitive derivation of this insight is presented in the appendix. The main argument rests on the observation that the relative contribution of a link to its two neighborhoods $\alpha_{ij}/\alpha_{ii} << 1$ and $\alpha_{ji}/\alpha_{jj} << 1$ can only be small simultaneously for large norms $\|\mathbf{a}\|^2 >> 1$ with multiple features that contribute to $\alpha_{ij}$. Yet, the norms $\|\mathbf{a}\|^2$ are constrained by the parameter initialization and cannot increase arbitrarily due to the derived conservation law.

To address this challenge, we modify the GAT architecture by GATE that learns separate attention parameters for the node and the neighborhood contribution. As its conservation law indicates, it can switch off neighborhood aggregation in the well-trainable parameter regime.

of feature and attention weights incoming to and outgoing from a neuron are preserved such that

**Theorem 4.3** (Structure of GATE gradients). *The gradients and parameters of GATE for layer $l \in [L-1]$ are conserved according to the following laws:*

$$\langle \mathbf{W}^l[i,:], \nabla_{\mathbf{W}^l[i,:]}\mathcal{L} \rangle - \langle \mathbf{a}_s^{l+1}[i], \nabla_{\mathbf{a}_s^{l+1}[i]}\mathcal{L} \rangle - \langle \mathbf{a}_t^{l+1}[i], \nabla_{\mathbf{a}_t^{l+1}[i]}\mathcal{L} \rangle = \langle \mathbf{W}^{l+1}[:,i], \nabla_{\mathbf{W}^{l+1}[:,i]}\mathcal{L} \rangle. \tag{7}$$

*and, if additional independent matrices $\mathbf{U}^l$ and $\mathbf{V}^l$ are trainable, it also holds*

$$\langle \mathbf{a}_s^l[i], \nabla_{\mathbf{a}_s^l[i]}\mathcal{L} \rangle + \langle \mathbf{a}_t^l[i], \nabla_{\mathbf{a}_t^l[i]}\mathcal{L} \rangle = \langle \mathbf{U}^l[i,:], \nabla_{\mathbf{U}^l[i,:]}\mathcal{L} \rangle + \langle \mathbf{V}^l[i,:], \nabla_{\mathbf{V}^l[i,:]}\mathcal{L} \rangle. \tag{8}$$

The proof is provided in the appendix. We utilize this theorem for two purposes. Firstly, it induces an initialization that enables at least the initial trainability of the network. Similarly to GAT (Mustafa & Burkholz, 2022), we initialize all attention parameters with zeros and the weight matrices with random orthogonal looks-linear structure in GATE. This also ensures that we have no initial inductive bias or preference for specific neighbor or node features. As an ablation, we also verify that the initialization of the attention parameters in GAT with zero can not, in fact, enable switching off neighborhood aggregation in GAT (see Fig. 5 in Appendix C).

Secondly, the conservation law leads to the insight that a GATE network is more easily capable of switching off neighborhood aggregation or node feature contributions in comparison with GAT.

**Insight 4.4** (GATE is able to switch off neighborhood aggregation.)**.** GATE can flexibly switch off neighborhood aggregation or node features in the well-trainable regime of the attention parameters.

This insight follows immediately from the related conservation law for GATE that shows that $a_t^l$ and $a_s^l$ can interchange the available budget for relative change among each other. Furthermore, the contribution of neighbors and the nodes are controlled separately so that the respective switch-off can be achieved with relatively small attention parameter norms that correspond to the well-trainable regime. To verify these insights in experiments, we next design synthetic data generators that can test the ability of GNNs to take graph data into account in a task-appropriate manner.

## 5 EXPERIMENTS

We validate the ability of GATE to perform the appropriate amount of neighborhood aggregation, as relevant for the given task and input graph, on both synthetic and real-world graphs. In order to gauge the amount of neighborhood aggregation, we study the distribution of $\alpha_{vv}$ values (over the nodes) at various epochs during training and layers in the network. This serves as a fair proxy since $\forall\, v \in \mathbb{V},\, \alpha_{vv} = 1 - \sum_{u \in \mathbb{N}(v), u \neq v} \alpha_{uv}$. Thus, $\alpha_{vv} = 1$ implies no neighborhood aggregation (i.e. only $\mathbf{h}_v$ is used) whereas $\alpha_{vv} = 0$ implies only neighborhood aggregation (i.e. $\mathbf{h}_v$ not is used). We defer a discussion of the experimental setup to Appendix B.

### 5.1 SYNTHETIC TEST BED

We construct the synthetic test bed as a node classification task for two types of problems: *self-sufficient* learning and *neighbor-dependent* learning. In the self-sufficient learning problem, complete label-relevant information is present in a node's own features. On the contrary, in the neighbor-dependent learning problem, label-relevant information is present in the node features of the $k$-hop neighbors. We discuss both cases in detail, beginning with the simpler self-sufficient case.

**Learning self-sufficient node labels**  In order to model this task exactly, we generate an Erdős–Rényi (ER) graph structure $G$ with $N = 1000$ nodes and edge probability $p = 0.01$. Node labels $y_v$ are assigned uniformly at random from $C = [2, 8]$ classes. Input node features $\mathbf{h}_v^0$ are generated as one-hot encoded node labels in both cases, i.e., $\mathbf{h}_v^0 = \mathbf{1}_{y_v}$. Nodes are divided randomly into train/validation/test split with a $2 : 1 : 1$ ratio.

We also use a real-world graph structure of the Cora dataset. Two cases using this graph structure are tested: i) using the original node labels consisting of 7 classes, and ii) randomized labels also of 7 classes. Input node features are generated as one-hot encoding of node labels in both cases. The standard train/validation/test splits of Cora are used.

As evident in Table 1, GAT is unable to perfectly learn this task whereas GATE easily achieves $100\%$ train and test accuracy, and often in fewer training epochs. Interestingly, a single layer GAT is able to almost, though not completely, switch off neighborhood aggregation (see Fig. 1) and achieve (near) perfect accuracy in the simpler cases. This is in line with our theoretical analysis (see Insight 4.2), as the norms of a single-layer model are not constrained and thus the attention parameters have more freedom to change. However, note that the accuracy of GAT worsens drastically along two dimensions simultaneously: i) an increase in the depth of the model (due to increased unnecessary aggregation), and ii) an increase in the complexity of the task (due to an increase in the number of classes in an ER graph and consequently in node neighborhoods).

In line with the homophilic nature of Cora, GAT achieves reasonably good accuracy when the original labels of the Cora graph structure are used as neighborhood aggregation is relatively less detrimental. Nevertheless, in the same case, GATE generalizes better than GAT with an increase in model depth. This indicates that over-smoothing, a major cause of performance degradation with model depth in GNNs, is also alleviated due to reduced neighborhood aggregations (see Fig. 1).

On the contrary, random labels pose a real challenge to GAT. Since the neighborhood features are fully uninformative about a node's label in the randomized case, aggregation over such a neighborhood distorts the fully informative features of the node itself. This impedes the GAT network from learning the task, as it is unable to effectively switch off aggregation (see Fig. 1), whereas GATE is able to adapt to the required level of neighborhood aggregation (i.e. none, in this case). In

the interest of space, we exclude here results for $\text{GATE}_S$ as similar performance and neighborhood aggregation patterns are observed as in GATE (see Fig. (8 in Appendix C).

Having established that GATE excels GAT in avoiding task-irrelevant neighborhood aggregation, it is also important to verify whether GATE can perform task-relevant neighborhood aggregation *when* required, and *as much as* required. We answer this question next by studying the behavior of GATE, in comparison to GAT, on a synthetically constructed neighbor-dependent learning problem.

Table 1: Self-sufficient learning: $S, C$ and $L$ denote graph structure, number of label classes, and number of network layers, respectively. Original (Orig.) and Randomized (Rand.) labels are used for the Cora structure. Most models achieve $100\%$ train accuracy, and entries marked with * refer to cases otherwise. $\text{GATE}_S$ also achieves $100\%$ train and test accuracy as GATE.

| $S$ | $C$ | $L$ | Test Acc.(%) @ Epoch of Min. Loss | | | Max Test Acc.(%) @ Epoch | | |
|---|---|---|---|---|---|---|---|---|
| | | | $\text{GAT}_S$ | GAT | GATE | $\text{GAT}_S$ | GAT | GATE |
| Cora | Orig.(7) | 1 | **99.1@215** | 97.7@166 | 99.0@127 | 100@300 | 100@548 | **100@196** |
| | | 2 | 93.4@218 | 94.5@158 | **99.6@35** | 94.4@187 | 94.6@156 | **100@40** |
| | | 5 | 85.9@92 | 85.5@72 | **98.4@36** | 88@10 | 88.5@16 | **99.7@51** |
| | Rand.(7) | 1 | 99.4@263 | 99.8@268 | **100@104** | 100@341 | 100@282 | **100@103** |
| | | 2 | 61.7@2088* | 52.8@341* | **99.9@36** | 67.0@2221 | 57.0@125 | **100@38** |
| | | 5 | 35.1@609 | 32.1@1299 | **99.9@23** | 37.4@327 | 36.7@12 | **100@42** |
| ER ($p = 0.01$) | Rand.(2) | 1 | 100@341 | **100@182** | 100@1313 | 100@340 | **100@181** | 100@1304 |
| | | 2 | 99.2@100 | 99.2@119 | **99.6@79** | 100@114 | 100@301 | **100@80** |
| | | 5 | 64.0@7778* | 99.6@239 | **100@45** | 75.6@163 | 99.6@224 | **100@45** |
| | Rand.(8) | 1 | 88.8@9578* | 98.4@3290 | **99.2@1755** | 90.4@9583 | 99.2@6727 | **100@2570** |
| | | 2 | 90.4@2459* | 94.8@2237 | **99.6@44** | 94.8@1973* | 97.6@6450 | **100@45** |
| | | 5 | 23.6@8152 | 26.0@8121 | **100@28** | 36.0@255* | 38.4@1011 | **100@27** |

**Learning neighbor-dependent node labels** In order to model this task, we generate an ER graph structure with $N = 1000$ nodes and edge probability $p = 0.01$. Input node features $\mathbf{h}_v^0 \in \mathbb{R}^d$ are sampled from a multivariate normal distribution $\mathcal{N}(\mathbf{0}_d, \mathbf{I}_d)$. For simplicity, $d = 2$.

This input graph $G$ is fed to a random GAT network $M_k$ with $k$ layers of width $d$. Note that this input graph $G$ has no self-loops on nodes (i.e. $v \notin \mathbb{N}(v)$). The parameters of $M_k$ are initialized with the standard Xavier (Glorot & Bengio, 2010) initialization. Thus, for each node $v$, the node embedding output by $M_k$, $\mathbf{h}_v^{M_k}$ is effectively a function $f$ of the $k$-hop neighboring nodes of node $v$ represented by a random GAT network. Let $\mathbb{N}_k(v)$ denote the set of $k$-hop neighbors of $v$ and $v \notin \mathbb{N}_k(v)$.

Finally, we run $K$-means clustering on the neighborhood aggregated representation of nodes $\mathbf{h}_v^{M_k}$ to divide nodes into $C$ clusters. For simplicity, we set $C = 2$. This clustering serves as the node labels (i.e. $y_v = \arg_{c \in [C]}(v \in c)$ for our node classification task. Thus, the label $y_v$ of a node $v$ to be learned is highly dependent on the input features of the neighboring nodes $\mathbf{h}_u^0 \in \mathbb{N}_k(v)$ rather than the node's own input features $\mathbf{h}_v^0$.

The generated input data and the real decision boundary for varying $k$ are shown in Fig. 2. Corresponding results in Table 2 and Fig. 3 exhibit that GATE can better detect the amount of necessary neighborhood aggregation than GAT. However, this task is more challenging than the previous one, and GATE too can not achieve perfect $100\%$ test accuracy. This could be attributed to data points close to the real decision boundary which is not well-defined and crisp (see Fig. 2).

## 5.2 REAL-WORLD DATA

We also analyze the behavior of GATE models on twelve real-world datasets (described in Table 4 in Appendix B) with varying homophily levels $\beta$ as defined in (Pei et al., 2020). Higher values of $\beta$ indicate higher homophily, i.e. similar nodes (with the same label) tend to be connected together.

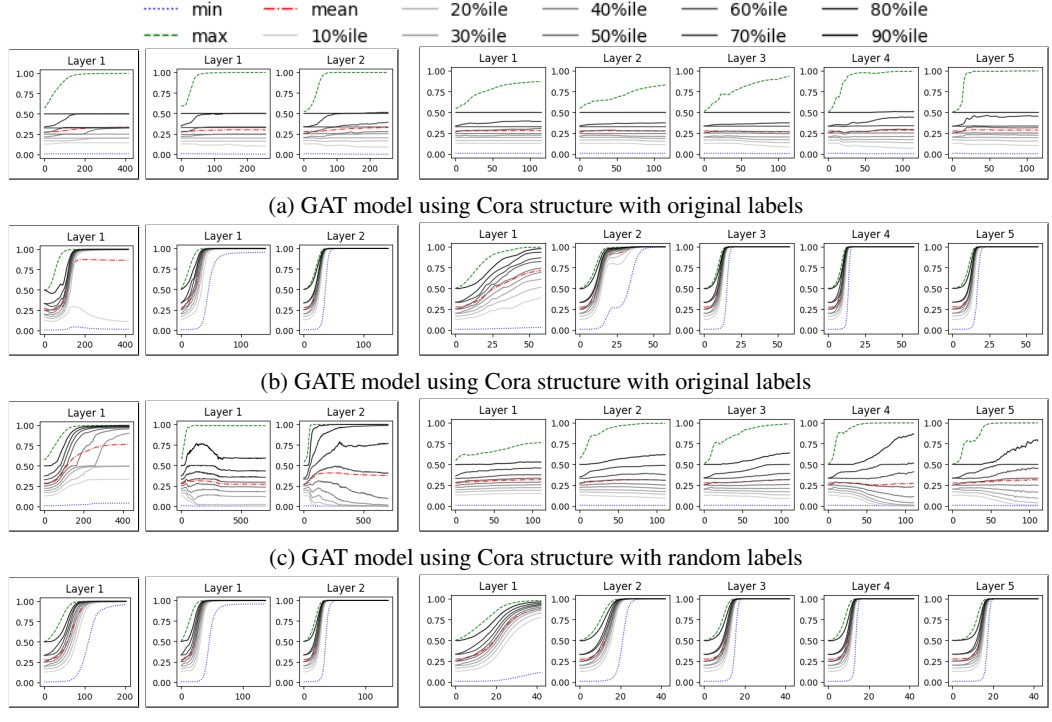

Figure 1: Distribution of $\alpha_{vv}$ against training epoch for self-sufficient learning problem, where input node features are a one-hot encoding of labels for 1, 2, and 5 layer models (left to right).

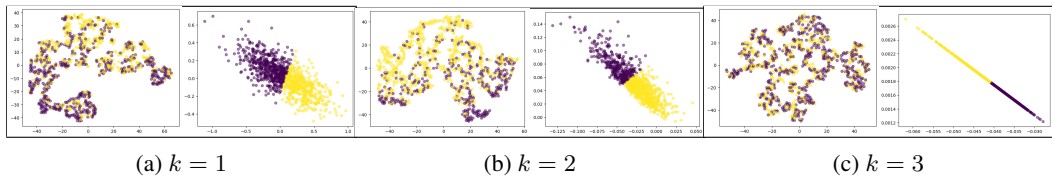

(a) $k = 1$        (b) $k = 2$        (c) $k = 3$

Figure 2: (a)-(c): Distribution of node labels of a synthetic dataset, with neighbor-dependent node labels, based on nodes' own random features (left) and neighbors' features aggregated $k$ times (right).

Table 3 reports the results for five datasets. Due to space limitation, we defer results on OGB (Hu et al., 2021) and large heterophilic (Platonov et al., 2023) datasets to Table 6 in appendix C. In particular, for heterophilic and OGB datasets, GATE outperforms GAT substantially by down-weighing connections to unrelated neighbors. We verify this by observing the neighborhood aggregation patterns of GATE in Fig. 4 that shows neighborhood aggregation is switched off in some layers by GATE. As expected, no layers switch off neighborhood aggregation in the GAT network (see Fig. (7) in Appendix C).

**Interpretable neighborhood aggregation** The distributions of $\alpha_{vv}$ in Fig. 4 across layers in a GATE model could be interpreted in terms of the inherent importance of input features of nodes relative to their neighborhood. For instance, in the case of Texas, GATE carries out little to no neighborhood aggregation in the first layer over input node features. Instead, aggregation is mainly done over node features transformed in earlier layers that effectuate non-linear feature learning as in perceptrons. However, in the case of Actor, GATE prefers most of the neighborhood aggregation to occur over the input node features, indicating that they are more informative for the task at hand.

Another interesting observation is that, when neighborhood aggregation takes place, the level of aggregation across all nodes, as indicated by the shape of $\alpha_{vv}$ distribution, varies over network layers. This is expected as different nodes need different levels of aggregation depending on where they are

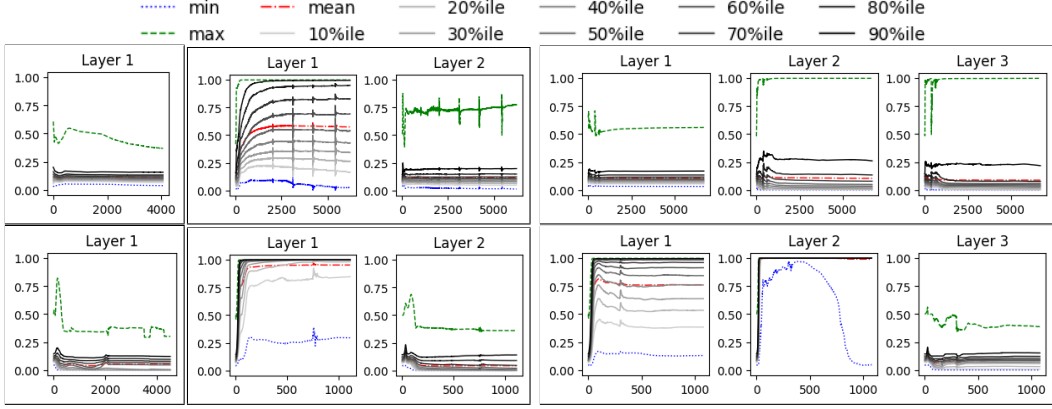

Figure 3: Distribution of $\alpha_{vv}$ against training epoch for the neighbor-dependent learning problem with $k = 1$. Rows: GAT (top) and GATE(bottom). Columns (left to right): 1, 2, and 3 layer models. While GAT is unable to switch off neighborhood aggregation, GATE allows most aggregation in mainly 1 layer of the 2 and 3 layer models. Similarly, we observe for $k = 3$, only 3 layers of the 4 and 5 layer models perform neighborhood aggregation (see Fig. 6 in Appendix C).

Table 2: Neighbor-dependent learning: $k$ and $L$ denote the number of aggregation steps of the random GAT used for label generation and the number of layers of the evaluated network, respectively. Entries marked with * identify models where $100\%$ train accuracy is not achieved. Underlined entries identify the model with the highest train accuracy at the epoch of max. test accuracy. This provides an insight into how similar the function represented by the trained model is to the function used to generate node labels. Higher training and test accuracy simultaneously indicate better learning. In this regard, the difference in train accuracy at max. test accuracy between GATE and GAT$_S$ or GAT is only $0.4$, $1.0$ and $0.6$ for the settings (k=1,L=3), (k=2,L=4), and (k=3,L=3), respectively.

| $k$ | $L$ | Test Acc. @ Epoch of Max. Train Acc. | | | Max Test Acc. @ Epoch | | |
| --- | --- | --- | --- | --- | --- | --- | --- |
| | | GAT$_S$ | GAT | GATE | GAT$_S$ | GAT | GATE |
| | 1 | 92.0@2082* | 91.2@6830* | **93.2@3712*** | 93.2@1421 | 92.0@9564 | **93.6@3511** |
| 1 | 2 | 89.6@8524* | 88.0@8935 | **91.2@942** | 91.6@5188 | 92.8@4198 | **95.6@111** |
| | 3 | 86.4@9180* | 88.8@997 | **92.8@618** | 91.2@6994 | 92.8@437 | **97.2@82** |
| | 2 | 88.8@6736* | **89.6@3907** | 88.8@467 | 93.2@151 | **93.2@95** | 92.0@105 |
| 2 | 3 | 82.0@7612 | 89.2@1950 | **91.6@370** | 91.6@1108 | 93.2@856 | **95.2@189** |
| | 4 | 84.8@4898 | 82.4@739 | **87.2@639** | 88.0@1744 | 88.4@423 | **90.4@447** |
| | 3 | 80.8@8670 | 80.4@737 | **85.2@391** | 86.4@1578 | 88.8@285 | **92.0@47** |
| 3 | 4 | 78.0@3012 | 80.4@767 | **89.6@480** | 86.8@1762 | 85.6@469 | **91.6@139** |
| | 5 | 80.0@6611 | 74.4@1701 | **86.0@447** | 85.6@921 | 83.6@1098 | **91.2@243** |

situated in the graph topology. For example, peripheral nodes would require more aggregation than central nodes to obtain a similar amount of information.

Therefore, as already observed with purposefully constructed synthetic data, GATE offers a more interpretable model than GAT in a real-world setting too. The distribution of the learned $\alpha_{uv}$ coefficients could reveal more meaningful information about the relative importance of node feature and graph structure information in the context of a given learning task, at the node level. Enabling this at the feature level could potentially further enhance the generalization and interpretability of GATE.

**Effect of depth** The ability of GATE to benefit from depth in terms of generalization is also demonstrated in the case of the Citeseer dataset. However, in general, GATE retains the same performance in the deeper models or suffers a smaller decrease in performance with an increase in

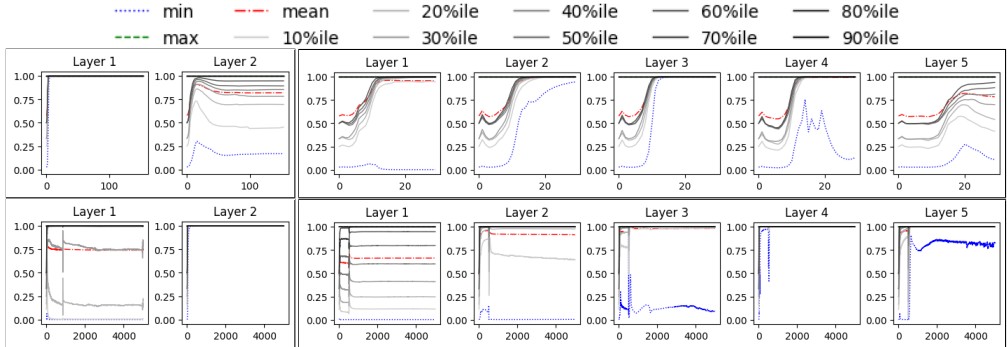

Figure 4: Distribution of $\alpha_{vv}$, against training epoch of 2-layer (left) and 5-layer (right) GATE networks for heterophilic datasets Texas (top) and Actor (bottom), across layers could be interpreted to indicate the inherent importance of raw node features relative to their neighborhoods.

depth, as compared to GAT. While the over-smoothing problem due to aggregation is peculiar to GNNs (and has been alleviated by addressing its root cause of unnecessary neighborhood aggregation to some extent (see Fig. 4), MLPs may also suffer from performance degradation with higher model depth. Standard techniques such as introducing skip connections in the network can also be used in combination with GATE to address this broader problem.

In this work, we focus our exposition on the neighborhood aggregation perspective of GATs alone. Therefore, we do not compare extensively with SOTA methods designed specifically for heterophilic datasets. However, in the general context of the problem complexity, we note that a 2-layer network of the baseline method for the heterophilic datasets, Geom-GCN (Pei et al., 2020), attains test accuracy (%) of $64.1$, $67.6$, and $31.6$ for Wisconsin, Texas, and Actor datasets, respectively, which is in line with that achieved by GATE.

Table 3: Test accuracy (%) of GAT and GATE models for network depth $L$ on real-world datasets with varying homophily levels $\beta$. Entries marked with * indicate models that achieve $100\%$ training accuracy and stable test accuracy. Otherwise, test accuracy at max. validation accuracy is reported.

| Data | $\beta$ | $L = 2$ | | $L = 5$ | | $L = 10$ | | $L = 20$ | |
|------|---------|---------|---------|---------|---------|---------|---------|---------|---------|
| | | GAT | GATE | GAT | GATE | GAT | GATE | GAT | GATE |
| Texas | .11 | 56.7* | **67.6*** | 51.4 | **67.6*** | 56.7* | **62.3*** | 59.4* | **64.9** |
| Wisc. | .21 | 62.7* | **70.5*** | 51.0 | **60.7*** | 45.1 | **58.8** | 47.1 | **60.7** |
| Actor | .24 | 27.1 | **31.6** | 25.4 | **29.2** | 25.3 | **27.9** | 24.5 | **29.4** |
| Cite. | .71 | 68.0 | **68.3** | 67.2 | **67.8** | 66.9 | **67.6** | 68.2 | **69.2** |
| Cora | .83 | 80.0 | **80.8** | 79.8 | **80.4** | 77.6 | **79.2** | 77.7 | **79.0** |

## 6   CONCLUSION

We experimentally illustrate a structural limitation of GAT that disables the architecture, in practice, to switch off task-irrelevant neighborhood aggregation. This obstructs GAT from achieving its intended potential. Based on insights from an existing conservation law of gradient flow dynamics in GAT, we can explain the source of its problem. To verify that we have identified the correct issue, we resolve it with a modification of GAT, which we call GATE, and derive the corresponding modified conservation law. GATE holds multiple advantages over GAT, as it can leverage the benefits of depth as in MLPs, offer interpretable, learned self-attention coefficients, and adapt the model to the necessary degree of neighborhood aggregation for a given task. Based on these properties, we argue that GAT is a suitable candidate to answer highly debated questions related to the importance of a given graph structure for standard tasks.

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

## A THEORETICAL DERIVATIONS

### A.1 DERIVATION OF INSIGHT 4.2

*Statement* (Restated Insight 4.2). GATs are challenged to switch off neighborhood aggregation during training, as this would require the model to enter a less trainable regime with large attention parameters $\|a\|^2 >> 1$.

We have to distinguish GATs with and without weight sharing in our analysis.

**GATs with weight sharing:**

To investigate the ability of a GAT to switch off neighborhood aggregation, let us focus on a link $(i, j)$ that should neither contribute to the feature transformation of $i$ nor $j$.

This implies that we need to find attention parameters $\mathbf{a}$ (and potentially feature transformations $W$) so that $\alpha_{ij}/\alpha_{ii} << 1$ with $\alpha_{ij}/\alpha_{ii} = \exp(e_{ij} - e_{ii})$. This implies that we require $e_{ij} - e_{ii} << 0$ and thus $\mathbf{a}^T \phi(\mathbf{W}(\mathbf{h}_i + \mathbf{h}_j)) - 2\mathbf{a}^T \phi(\mathbf{W}(\mathbf{h}_i)) << 0$.

Since we also require $\alpha_{ij}/\alpha_{jj} << 1$, it follows from adding both inequalities that $\mathbf{a}^T [\phi(\mathbf{W}(\mathbf{h}_i + \mathbf{h}_j)) - (\phi(\mathbf{W}\mathbf{h}_i) + \phi(\mathbf{W}\mathbf{h}_j))] << 0$.

This inequality can only be fulfilled if there exists at least one feature $f$ for which

$$\Delta_{fij} := a[f] [\phi(\mathbf{W}[f,:](\mathbf{h}_i + \mathbf{h}_j)) - (\phi(\mathbf{W}[f,:]\mathbf{h}_i) + \phi(\mathbf{W}[f,:]\mathbf{h}_i))]$$

fulfills $\Delta_{fij} << 0$. Yet, note that if both $\phi(\mathbf{W}[f,:]\mathbf{h}_i)$ and $\phi(\mathbf{W}[f,:]\mathbf{h}_j)$ are positive or both are negative, we just get $\Delta_{fij} = 0$ because of the definition of a LeakyReLU. Thus, there must exist at least one feature $f$ so that without loss of generality $\phi(\mathbf{W}[f,:]\mathbf{h}_i) < 0$ and $\phi(\mathbf{W}[f,:]\mathbf{h}_j) > 0$.

It follows that if $a[f] > 0$ that

$$0 > a[f]\phi(\mathbf{W}[f,:]\mathbf{h}_i) >> a[f](\phi(\mathbf{W}[f,:](\mathbf{h}_i + \mathbf{h}_j)) - \phi(\mathbf{W}[f,:]\mathbf{h}_j))$$
$$> a[f](\phi(\mathbf{W}[f,:](\mathbf{h}_i + \mathbf{h}_j)) - 2\phi(\mathbf{W}[f,:]\mathbf{h}_j))$$

also receives a negative contribution that makes $\alpha_{ij}/\alpha_{jj}$ smaller. Yet, what happens to $\alpha_{ij}/\alpha_{ii}$? By distinguishing two cases, namely $\mathbf{W}[f,:](\mathbf{h}_i + \mathbf{h}_j) > 0$ or $\mathbf{W}[f,:](\mathbf{h}_i + \mathbf{h}_j) < 0$ and computing

$$a[f][\phi(\mathbf{W}(\mathbf{h}_i + \mathbf{h}_j)) - 2\phi(\mathbf{W}[f,:]\mathbf{h}_j)] > 0$$

we find the feature contribution to be positive.

If $a[f] < 0$, then

$$0 > a[f]\phi(\mathbf{W}[f,:]\mathbf{h}_j) >> a[f](\phi(\mathbf{W}[f,:](\mathbf{h}_i + \mathbf{h}_j)) - \phi(\mathbf{W}[f,:]\mathbf{h}_i))$$
$$> a[f](\phi(\mathbf{W}[f,:](\mathbf{h}_i + \mathbf{h}_j)) - 2\phi(\mathbf{W}[f,:]\mathbf{h}_i))$$

and $\alpha_{ij}/\alpha_{jj}$ is reduced. Similarly, we can derive that at the same time $\alpha_{ij}/\alpha_{ii}$ is increased, however.

This implies that any feature that contributes to reducing $\Delta_{fij}$ automatically increases one feature while it increases another. We therefore need multiple features $f$ to contribute to reducing either $\alpha_{ij}/\alpha_{ii}$ or $\alpha_{ij}/\alpha_{jj}$ to compensate for other increases.

This implies, in order to switch off neighborhood aggregation, we would need a high dimensional space of features that cater to switching off specific links without strengthening others. Furthermore, they would need large absolute values of $a[f]$ and norms of $\mathbf{W}[f,:]$ or exploding feature vectors $\mathbf{h}$ to achieve this.

Yet, all these norms are constrained by the derived conservation law and therefore prevent learning a representation that switches off full neighborhoods.

**GATs without weight sharing:**

The flow of argumentation without weight sharing is very similar to the one above with weight sharing. Yet, we have to distinguish more cases.

Similarly to before, we require $\alpha_{ij}/\alpha_{jj} << 1$ and $\alpha_{ji}/\alpha_{ii} << 1$. It follows from adding both related inequalities that

$$\mathbf{a}^T\left[\phi\left(\mathbf{W}_s\mathbf{h}_i+\mathbf{W}_t\mathbf{h}_j\right)+\phi\left(\mathbf{W}_s\mathbf{h}_j+\mathbf{W}_t\mathbf{h}_i\right)-\phi\left(\left(\mathbf{W}_s+\mathbf{W}_t\right)\mathbf{h}_i\right)-\phi\left(\left(\mathbf{W}_s+\mathbf{W}_t\right)\mathbf{h}_j\right)\right]<<0.$$

This implies that for at least one feature $f$, we require

$$
\begin{aligned}
a[f][\phi\left(\mathbf{W}_s[f,:]\mathbf{h}_i+\mathbf{W}_t[f,:]\mathbf{h}_j\right)+\phi\left(\mathbf{W}_s[f,:]\mathbf{h}_j+\mathbf{W}_t[f,:]\mathbf{h}_i\right)\\
-\phi\left(\left(\mathbf{W}_s[f,:]+\mathbf{W}_t[f,:]\right)\mathbf{h}_i\right)-\phi\left(\left(\mathbf{W}_s[f,:]+\mathbf{W}_t[f,:]\right)\mathbf{h}_j\right)]<<0.
\end{aligned}
\tag{9}
$$

Again, our goal is to show that this feature automatically decreases the contribution of one feature while it increases another. As argued above, switching off neighborhood aggregation would therefore need a high dimensional space of features that cater to switching off specific links without strengthening others. Furthermore, they would need large absolute values of $a[f]$ and norms of $\mathbf{W}[f,:]$ or exploding feature vectors $\mathbf{h}$ to achieve this. Our derived norm constraints, however, prevent learning such a model representation.

Concretely, without loss of generality, we therefore have to show that if

$$a[f][\phi\left(\mathbf{W}_s[f,:]\mathbf{h}_i+\mathbf{W}_t[f,:]\mathbf{h}_j\right)-\phi\left(\left(\mathbf{W}_s[f,:]+\mathbf{W}_t[f,:]\right)\mathbf{h}_j\right)<0, \tag{10}$$

at the same time, we receive

$$a[f][\phi\left(\mathbf{W}_s[f,:]\mathbf{h}_j+\mathbf{W}_t[f,:]\mathbf{h}_i\right)-\phi\left(\left(\mathbf{W}_s[f,:]+\mathbf{W}_t[f,:]\right)\mathbf{h}_i\right)>0, \tag{11}$$

(or vice versa).

In principle, we have to show this for 16 different cases of pre-activation sign configurations for the four terms in Eq. (9). Yet, since the argument is symmetric with respect to exchanging $i$ and $j$, only 8 different cases remain. Two trivial cases are identical signs for all four terms. These are excluded, as the left hand side (LHS) of Eq. (9) would become zero and thus not contribute to our goal to switch off neighborhood aggregation. In the following, we will discuss the remaining six cases. Please note that for the remainder of this derivation $\alpha > 0$ denotes the slope of the leakyReLU and not the attention weights $\alpha_{ij}$.

**Case $(+-++)$:** Let us assume that $\mathbf{W}_s[f,:]\mathbf{h}_i+\mathbf{W}_t[f,:]\mathbf{h}_j > 0$, $\mathbf{W}_s[f,:]\mathbf{h}_j+\mathbf{W}_t[f,:]\mathbf{h}_i < 0$, $(\mathbf{W}_s[f,:]+\mathbf{W}_t[f,:])\mathbf{h}_i > 0$, and $(\mathbf{W}_s[f,:]+\mathbf{W}_t[f,:])\mathbf{h}_j > 0$.

From this assumption and the fact that $\phi$ is a leakyReLU it follows that the LHS of Eq. (9) becomes: $a[f][\phi\left(\mathbf{W}_s[f,:]\mathbf{h}_i+\mathbf{W}_t[f,:]\mathbf{h}_j\right)+\phi\left(\mathbf{W}_s[f,:]\mathbf{h}_j+\mathbf{W}_t[f,:]\mathbf{h}_i\right)-\phi\left(\left(\mathbf{W}_s[f,:]+\mathbf{W}_t[f,:]\right)\mathbf{h}_i\right)-\phi\left(\left(\mathbf{W}_s[f,:]+\mathbf{W}_t[f,:]\right)\mathbf{h}_j\right)] = a[f](\alpha-1)[\mathbf{W}_s[f,:]\mathbf{h}_j+\mathbf{W}_t[f,:]\mathbf{h}_i]$. Since $\alpha - 1 < 0$ and $[\mathbf{W}_s[f,:]\mathbf{h}_j+\mathbf{W}_t[f,:]\mathbf{h}_i] < 0$ according to our assumption, Eq. (9) demands $a[f] < 0$. To switch off neighborhood aggregation, we would need to be able to make the LHS of Eq. (10) and Eq. (11) Eq. (11) negative. Yet, a negative $a[f]$ leads to a positive LHS of Eq. (11). Thus, the assumed sign configuration cannot support switching off neighborhood aggregation.

**Case $(+---)$:** Let us assume that $\mathbf{W}_s[f,:]\mathbf{h}_i+\mathbf{W}_t[f,:]\mathbf{h}_j > 0$, $\mathbf{W}_s[f,:]\mathbf{h}_j+\mathbf{W}_t[f,:]\mathbf{h}_i < 0$, $(\mathbf{W}_s[f,:]+\mathbf{W}_t[f,:])\mathbf{h}_i < 0$, and $(\mathbf{W}_s[f,:]+\mathbf{W}_t[f,:])\mathbf{h}_j < 0$.

The LHS of Eq. (9) becomes $a[f](1-\alpha)[\mathbf{W}_s[f,:]\mathbf{h}_i+\mathbf{W}_t[f,:]\mathbf{h}_j]$, which demands $a[f] < 0$. Accordingly, the LHS of Eq. (10) is clearly negative, while the LHS of Eq. (11) is $a[f]\alpha\mathbf{W}_s[f,:](\mathbf{h}_j-\mathbf{h}_i) > 0$. The last inequality follows from our assumptions that imply $\mathbf{W}_s[f,:]\mathbf{h}_j < \mathbf{W}_s[f,:]\mathbf{h}_i$ by combining the assumptions $(\mathbf{W}_s[f,:]+\mathbf{W}_t[f,:])\mathbf{h}_j < 0$ and $\mathbf{W}_s[f,:]\mathbf{h}_i+\mathbf{W}_t[f,:]\mathbf{h}_j > 0$. Again, this result implies that the considered sign configuration does not support switching off neighborhood aggregation.

**Case $(++ +-)$:** Let us assume that $\mathbf{W}_s[f,:]\mathbf{h}_i+\mathbf{W}_t[f,:]\mathbf{h}_j > 0$, $\mathbf{W}_s[f,:]\mathbf{h}_j+\mathbf{W}_t[f,:]\mathbf{h}_i > 0$, $(\mathbf{W}_s[f,:]+\mathbf{W}_t[f,:])\mathbf{h}_i > 0$, and $(\mathbf{W}_s[f,:]+\mathbf{W}_t[f,:])\mathbf{h}_j < 0$.

The LHS of Eq. (9) becomes $a[f](1-\alpha)[\mathbf{W}_s[f,:]\mathbf{h}_j+\mathbf{W}_t[f,:]\mathbf{h}_j]$, which demands $a[f] > 0$. Accordingly, the LHS of Eq. (10) becomes positive, which hampers switching-off neighborhood aggregation as discussed.

**Case $(---+)$:** Let us assume that $\mathbf{W}_s[f,:]\mathbf{h}_i+\mathbf{W}_t[f,:]\mathbf{h}_j < 0$, $\mathbf{W}_s[f,:]\mathbf{h}_j+\mathbf{W}_t[f,:]\mathbf{h}_i < 0$, $(\mathbf{W}_s[f,:]+\mathbf{W}_t[f,:])\mathbf{h}_i < 0$, and $(\mathbf{W}_s[f,:]+\mathbf{W}_t[f,:])\mathbf{h}_j > 0$.

The LHS of Eq. (9) becomes $a[f](\alpha - 1)[\mathbf{W}_s[f,:]\mathbf{h}_j + \mathbf{W}_t[f,:]\mathbf{h}_j]$, which demands $a[f] > 0$. Accordingly, the LHS of Eq. (10) becomes clearly negative. However, the LHS of Eq. (11) is positive, as $a[f]\alpha\mathbf{W}_s[f,:](\mathbf{h}_j - \mathbf{h}_i) > 0$.

The last inequality follows from our assumptions that imply $\mathbf{W}_s[f,:]\mathbf{h}_j > \mathbf{W}_s[f,:]\mathbf{h}_i$ by combining the assumptions $(\mathbf{W}_s[f,:] + \mathbf{W}_t[f,:])\mathbf{h}_j > 0$ and $\mathbf{W}_s[f,:]\mathbf{h}_i + \mathbf{W}_t[f,:]\mathbf{h}_j < 0$. Again, this analysis implies that the considered sign configuration does not support switching off neighborhood aggregation.

**Case** $(+ - + -)$: Let us assume that $\mathbf{W}_s[f,:]\mathbf{h}_i + \mathbf{W}_t[f,:]\mathbf{h}_j > 0$, $\mathbf{W}_s[f,:]\mathbf{h}_j + \mathbf{W}_t[f,:]\mathbf{h}_i < 0$, $(\mathbf{W}_s[f,:] + \mathbf{W}_t[f,:])\mathbf{h}_i > 0$, and $(\mathbf{W}_s[f,:] + \mathbf{W}_t[f,:])\mathbf{h}_j < 0$.

According to our assumptions the LHS of Eq. (10) can only be negative if $a[f] < 0$. Yet, the LHS of Eq. (11) can only be negative if $a[f] > 0$. Thus, this case clearly cannot contribute to switching off neighborhood aggregation.

**Case** $(+ - - +)$: Let us assume that $\mathbf{W}_s[f,:]\mathbf{h}_i + \mathbf{W}_t[f,:]\mathbf{h}_j > 0$, $\mathbf{W}_s[f,:]\mathbf{h}_j + \mathbf{W}_t[f,:]\mathbf{h}_i < 0$, $(\mathbf{W}_s[f,:] + \mathbf{W}_t[f,:])\mathbf{h}_i < 0$, and $(\mathbf{W}_s[f,:] + \mathbf{W}_t[f,:])\mathbf{h}_j > 0$.

Eq. (9) becomes $a[f](1 - \alpha)\mathbf{W}_s[f,:](\mathbf{h}_i - \mathbf{h}_j) < 0$. At the same time, the LHS of Eq. (10) simplifies to $a[f]\mathbf{W}_s[f,:](\mathbf{h}_i - \mathbf{h}_j)$ and the LHS of Eq. (11) is $a[f]\alpha\mathbf{W}_s[f,:](\mathbf{h}_j - \mathbf{h}_i) > 0$.

Hence, a negative Eq. (9) leads to a positive Eq. (11). Accordingly, the last possible sign configuration also does not support switching off neighborhood aggregation, which concludes our derivation.

## A.2 PROOF OF THEOREM 4.3

*Statement* (Restated Theorem 4.3). The gradients and parameters of GATE for layer $l \in [L - 1]$ are conserved according to the following laws:

$$\langle W^l[i,:], \nabla_{W^l[i,:]}\mathcal{L}\rangle = \langle W^{l+1}[:,i], \nabla_{W^{l+1}[:,i]}\mathcal{L}\rangle + \langle a_s^{l+1}[i], \nabla_{a_s^{l+1}[i]}\mathcal{L}\rangle + \langle a_t^{l+1}[i], \nabla_{a_t^{l+1}[i]}\mathcal{L}\rangle. \quad (12)$$

and, if additional independent matrices $\mathbf{U}^l$ and $\mathbf{V}^l$ are trainable, it also holds

$$\langle a_s^l[i], \nabla_{a_s^l[i]}\mathcal{L}\rangle + \langle a_t^l[i], \nabla_{a_t^l[i]}\mathcal{L}\rangle = \langle U^l[i,:], \nabla_{U^l[i,:]}\mathcal{L}\rangle + \langle V^l[i,:], \nabla_{V^l[i,:]}\mathcal{L}\rangle. \quad (13)$$

The proof is analogous to the derivation of Theorem 2.2 by (Mustafa & Burkholz, 2022) that is restated in this work as Theorem 4.1. For ease, we replicate their notation and definitions here.

*Statement* (Rescale invariance: Def 5.1 by Mustafa & Burkholz (2022)). The loss $\mathcal{L}(\theta)$ is rescale-invariant with respect to disjoint subsets of the parameters $\theta_1$ and $\theta_2$ if for every $\lambda > 0$ we have $\mathcal{L}(\theta) = \mathcal{L}((\lambda\theta_1, \lambda^{-1}\theta_2, \theta_d))$, where $\theta = (\theta_1, \theta_2, \theta_d)$.

*Statement* (Gradient structure due to rescale invariance Lemma 5.2 in Mustafa & Burkholz (2022)). The rescale invariance of $\mathcal{L}$ enforces the following geometric constraint on the gradients of the loss with respect to its parameters:

$$\langle \theta_1, \nabla_{\theta_1}\mathcal{L}\rangle - \langle \theta_2, \nabla_{\theta_2}\mathcal{L}\rangle = 0. \quad (14)$$

We first consider the simpler case of GATE$_S$, i.e. $W = U = V$

**Theorem A.1** (Structure of GATE$_S$ gradients). *The gradients and parameters of GATE$_S$ for layer $l \in [L - 1]$ are conserved according to the following laws:*

$$\langle W^l[i,:], \nabla_{W^l[i,:]}\mathcal{L}\rangle = \langle W^{l+1}[:,i], \nabla_{W^{l+1}[:,i]}\mathcal{L}\rangle + \langle a_s^l[i], \nabla_{a_s^l[i]}\mathcal{L}\rangle + \langle a_t^l[i], \nabla_{a_t^l[i]}\mathcal{L}\rangle. \quad (15)$$

Following a similar strategy to (Mustafa & Burkholz, 2022), we identify rescale invariances for every neuron $i$ at layer $l$ that induce the stated gradient structure.

Given the following definition of disjoint subsets $\theta_1$ and $\theta_2$ of the parameter set $\theta$, associated with neuron $i$ in layer $l$,

$$\theta_1 = \{x | x \in W^l[i,:]\}$$
$$\theta_2 = \{w | w \in W^{l+1}[:,i]\} \cup \{a_s^l[i]\} \cup \{a_t^l[i]\}$$

We show that the loss of $\text{GATE}_S$ remains invariant for any $\lambda > 0$.

The only components of the network that potentially change under rescaling are $h_u^l[i]$, $h_v^{l+1}[j]$, and $\alpha_{uv}^l$.

The scaled network parameters are denoted with a tilde as $\tilde{a_s}^l[i] = \lambda^{-1}a_s^l[i]$, $\tilde{a_t}^l[i] = \lambda^{-1}a_t^l[i]$, and $\tilde{W}^l[i,j] = \lambda W^l[i,j]$, and the corresponding networks components scaled as a result are denoted by $\tilde{h}_u^l[i]$, $\tilde{h}_v^{l+1}[k]$, and $\tilde{\alpha}_{uv}^l$.

We show that the parameters of upper layers remain unaffected, as $\tilde{h}_v^{l+1}[k]$ coincides with its original non-scaled variant $\tilde{h}_v^{l+1}[k] = h_v^{l+1}[k]$.

Also recall Eq. (4) for $W = U = V$ as:

$$e_{uv}^l = ((1 - q_{uv})a_s^l + (q_{uv})a_t^l)^\top \cdot \phi(W^l h_u^{l-1} + W^l h_v^{l-1})$$

where $q_{uv} = 1$ if $u = v$ and $q_{uv} = 0$ if $u \neq v$.

For simplicity, we rewrite this as:

$$e_{uv,u\neq v}^l = (a_s^l)^\top \cdot \phi(W^l h_u^{l-1} + W^l h_v^{l-1}) \tag{16}$$

$$e_{uv,u=v}^l = (a_t^l)^\top \cdot \phi(W^l h_u^{l-1} + W^l h_v^{l-1}) \tag{17}$$

We show that

$$\tilde{\alpha}_{uv}^l = \frac{\exp(\tilde{e}_{uv}^l)}{\sum_{u' \in \mathcal{N}(v)} \exp(\tilde{e}_{uv}^l)} = \alpha_{uv}^l , \quad \text{because} \tag{18}$$

$$\tilde{e}_{uv,u\neq v}^l = e_{uv,u\neq v}^l , \quad \text{and} \quad \tilde{e}_{uv,u=v}^l = e_{uv,u=v}^l \tag{19}$$

which follows from the positive homogeneity of $\phi$ that allows

$$\tilde{e}_{uv,u=v}^l = \lambda^{-1}a_s^l[i]\phi(\sum_j^{n_{l-1}} \lambda W^l[i,j](h_u^{l-1}[j] + h_v^{l-1}[j])$$

$$+ \sum_{i'\neq i}^{n_l} a_s^l[i']\phi(\sum_j^{n_{l-1}} W^l[i',j](h_u^{l-1}[j] + h_v^{l-1}[j]) \tag{20}$$

$$= \lambda^{-1}\lambda a_s^l[i]\phi(\sum_j^{n_{l-1}} W^l[i,j](h_u^{l-1}[j] + h_v^{l-1}[j])$$

$$+ \sum_{i'\neq i}^{n_l} a_s^l[i']\phi(\sum_j^{n_{l-1}} W^l[i',j](h_u^{l-1}[j] + h_v^{l-1}[j]) \tag{21}$$

$$= e_{uv,u\neq v}^l. \tag{22}$$

and similarly,

$$\tilde{e}^l_{uv,u=v} = \lambda^{-1} a^l_t[i] \phi \left( \sum_j^{n_{l-1}} \lambda W^l[i,j](h^{l-1}_u[j] + h^{l-1}_v[j]) \right.$$

$$+ \sum_{i' \neq i}^{n_l} a^l_t[i'] \phi \left( \sum_j^{n_{l-1}} W^l[i',j](h^{l-1}_u[j] + h^{l-1}_v[j]) \right) \tag{23}$$

$$= \lambda^{-1} \lambda a^l_t[i] \phi \left( \sum_j^{n_{l-1}} W^l[i,j](h^{l-1}_u[j] + h^{l-1}_v[j]) \right)$$

$$+ \sum_{i' \neq i}^{n_l} a^l_t[i'] \phi \left( \sum_j^{n_{l-1}} W^l[i',j](h^{l-1}_u[j] + h^{l-1}_v[j]) \right) \tag{24}$$

$$= e^l_{uv,u=v}. \tag{25}$$

Since $\tilde{\alpha}^l_{uv} = \alpha^l_{uv}$, it follows that

$$\tilde{h}^l_u[i] = \phi_1 \left( \sum_{z \in \mathcal{N}(u)} \alpha^l_{zu} \sum_j^{n_{l-1}} \lambda W^l[i,j] h^{l-1}_z[j] \right)$$

$$= \lambda \phi_1 \left( \sum_{z \in \mathcal{N}(u)} \alpha^l_{zu} \sum_j^{n_{l-1}} W^l[i,j] h^{l-1}_z[j] \right)$$

$$= \lambda h^l_u[i].$$

In the next layer, we therefore have

$$\tilde{h}^{l+1}_v[k] = \phi_1 \left( \sum_{u \in \mathcal{N}(v)} \alpha^{l+1}_{uv} \sum_i^{n_l} \lambda^{-1} W^{l+1}[k,i] \tilde{h}^l_u[i] \right)$$

$$= \phi_1 \left( \sum_{u \in \mathcal{N}(v)} \alpha^{l+1}_{uv} \sum_i^{n_l} \lambda^{-1} W^{l+1}[k,i] \lambda h^l_u[i] \right)$$

$$= \phi_1 \left( \sum_{u \in \mathcal{N}(v)} \alpha^{l+1}_{uv} \sum_i^{n_l} W^{l+1}[k,i] h^l_u[i] \right)$$

$$= h^{l+1}_v[k].$$

Thus, the output node representations of the network remain unchanged, and the loss $\mathcal{L}$ is rescale-invariant.

Next consider the case that $W^l$, $U^l$, and $V^l$ are independent matrices. Similarly to the previous reasoning, we see that if we scale $\tilde{W}^l[i,:] = W^l[i,:]\lambda$, then also scaling $\tilde{W}^{l+1}[:,i] = W^{l+1}[:,i]\lambda^{-1}$ and $\tilde{a}^{l+1}_s[i] = a^{l+1}_s[i]\lambda^{-1}$ and $\tilde{a}^{l+1}_t[i] = a^{l+1}_t[i]\lambda^{-1}$ will keep the GATE layer unaltered.

In this case, we obtain an additional rescaling relationship between $a^l_s$, $a^l_t$ and $U^l$, $V^l$. A rescaling of the form $\tilde{a}^l_s[i] = \lambda^{-1} a^l_s[i]$, $\tilde{a}^l_t[i] = \lambda^{-1} a^l_t[i]$ could be compensated by $\tilde{U}^l[i,:] = U^l[i,:]\lambda$ and $\tilde{V}^l[i,:] = V^l[i,:]\lambda$. It follows immediately that $\tilde{e}_{uv} = e_{uv}$.

## A.3 DERIVATION OF INSIGHT 4.4

Following the analysis in A.1, in contrast to GAT, $\alpha_{ij}/\alpha_{ii} << 1$ can be easily realized in GATE with $a_s[f] < 0$ and $a_t[f] > 0$ for all or only a subset of the features. Note that for the non-weight-sharing case, $\mathbf{U}$ and $\mathbf{V}$ in GATE would simply correspond to $\mathbf{W}_s$ and $\mathbf{W}_t$, respectively, in GATE and the same line of reasoning holds. Large norms are usually not required to create a notable difference in size between $e_{ii}$ and $e_{ij}$.

## B  EXPERIMENTAL SETTINGS

We vary the depth of GAT and GATE networks in our experiments, but keep the hidden layer width fixed to 64 in all cases. For $\text{GAT}_S$ and GAT networks, we substitute $\phi$ in Eq. (3) with LeakyReLU as defined in the standard architecture. For GATE, we substitute $\phi$ in Eq. (4) with ReLU in order to be able to interpret the sign of $\mathbf{a_s}$ and $\mathbf{a_t}$ parameters as contributing positively or negatively to neighborhood aggregation. For synthetic and real-world data, a maximum of 10000 and 5000 epochs are run, respectively, using the Adam optimizer. In order to isolate the effect of the architecture and study the parameter dynamics during training as best as possible, we do not use any additional elements such as weight decay and dropout regularization. We also do not perform any hyperparameter optimization. However, the learning rate is adjusted for different real-world datasets to enable stable training of models as specified in Table 5. Nevertheless, for a fair comparison, the same learning rate is used for a given problem across all architectures. For all synthetic data, a learning rate of 0.005 is used. Real-world datasets use their standard train/test/validation splits. The feature transformation parameters, i.e., $\mathbf{W}, \mathbf{U}$, and $\mathbf{V}$ are initialized randomly as looks-linear orthogonal (Burkholz & Dubatovka, 2019). The parameters $\mathbf{a}$ in $\text{GAT}_S$ and GAT use Xavier initialization (Glorot & Bengio, 2010), as is the standard. In GATE, $\mathbf{a}_s$ and $\mathbf{a}_t$ are initialized to 0 in order to initially give equal weights to the features of a node itself and its neighboring nodes.

Table 4: Details of the real-world datasets used in experiments.

| Dataset | # Nodes | # Edges | # Features | # Classes | # Train | # Validate | # Test |
|---------|---------|---------|------------|-----------|---------|------------|--------|
| Cora | 2708 | 10556 | 1433 | 7 | 140 | 500 | 1000 |
| Citeseer | 3327 | 9104 | 3703 | 6 | 120 | 500 | 1000 |
| Actor | 7600 | 26659 | 932 | 5 | 3648 | 2432 | 1520 |
| Texas | 183 | 279 | 1703 | 5 | 87 | 59 | 37 |
| Wisconsin | 251 | 450 | 1703 | 5 | 120 | 80 | 51 |

Table 5: Learning rate used in experiments on real-world datasets. $L$ is the number of network layers.

| $L$ | Cora | Citeseer | Wisconsin | Texas | Actor |
|-----|------|----------|-----------|-------|-------|
| 2 | 0.005 | 0.005 | 0.01 | 0.01 | 0.005 |
| 5 | 0.005 | 0.005 | 0.01 | 0.01 | 0.005 |
| 10 | 0.0005 | 0.0001 | 0.005 | 0.0005 | 0.005 |
| 20 | 0.0005 | 0.0001 | 0.005 | 0.0005 | 0.005 |

## C  ADDITIONAL RESULTS

### C.1  LARGE REAL-WORLD DATASETS

We also evaluate GATE in comparison to GAT on OGB datasets, Arxiv, and Products Hu et al. (2021), and larger heterophilic datasets proposed by Platonov et al. (2023). The results reported in Table 6 show that GATE substantially improves the performance on the larger OGB dataset Arxiv and theheterophilic dataset, tolokers, by allowing the network to leverage depth without over-smoothing. The greatest gain in performance is observed for the roman-empire dataset. Except for the amazon-ratings dataset (where the performance is comparable), GATE outperforms GAT on all datasets by a large margin.

Table 6: Test accuracy reported for OGB datasets (arxiv, products), roman-empire, and amazon-ratings. AUC-ROC reported for binary classification datasets minesweeper, tolokers, and question datasets following (Platonov et al., 2023). *As reported by (Brody et al., 2022) for 3 layer GAT after hyperparameter search for the number of layers $L$ in $\{2, 3, 6\}$.

| Dataset | $L$ | GATE | GAT |
|---|---|---|---|
| minesweeper | 5 | **.676** | .505 |
| tolokers | 5 | .638 | .638 |
| | 10 | **.692** | .616 |
| questions | 5 | **.643** | .548 |
| roman-empire | 5 | **.759** | .290 |
| amazon-ratings | 5 | .454 | **.459** |
| ogb-arxiv | 3 | .773 | .719* |
| | 5 | .784 | - |
| | 8 | **.795** | - |
| ogb-products | 3 | .861 | .806* |
| | 5 | **.863** | - |

### C.2  ABLATIONS

We conduct two ablations. Firstly, in Figure 5, we evaluate a GAT with initial values of attention parameters set to 0, and show that this setting is not what enables neighborhood aggregation in GATE. Secondly, in Table 7, we compare the weight-sharing and non-weight-sharing cases of GATE on real-world datasets. GATE usually outperforms, particularly at deeper layers. Although GATE is more parameterized than GAT, it usually requires fewer training epochs and generalizes better, in addition to other advantages over GAT as discussed in the paper.

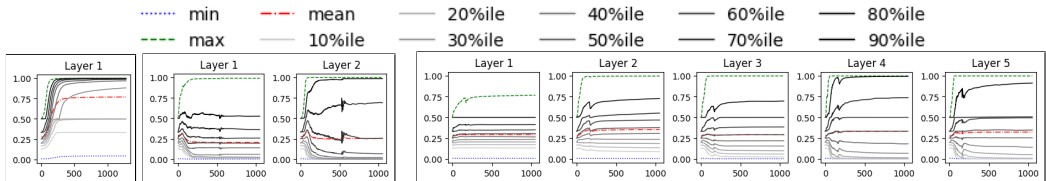

Figure 5: Distribution of $\alpha_{vv}$ against training epoch for self-sufficient learning problem using the Cora structure with random labels, where input node features are a one-hot encoding of node labels for GAT with attention parameters **a** initialized to zero. Left to right: 1, 2 and 5 layer models that achieve test accuracy of 100%, 52.7%, and 36.2%, respectively, which is similar to the results obtained by standard Xavier initialization of attention parameters in GAT. This ablation shows that setting the initial value of attention parameters $a_s$ and $a_t$ in GATE to zero is, in fact, not what enables neighborhood aggregation but rather the separation of $a$ into $a_s$ and $a_t$ as discussed in Insight 4.4.

Table 7: Test accuracy (%) of GATE$_S$ models for network depth $L$ on real data. For comparison, we restate the results for GATE from Table 3. Entries marked with * indicate models that achieve 100% training accuracy and stable test accuracy. Otherwise, test accuracy at max. validation accuracy is reported. The same learning rates were used as in Table 5 except for the 20-layer network for Wisconsin dataset, which used a learning rate of 0.0001 instead. Generally, except for Wisconsin, where GATE$_S$ outperforms GATE (and thus GAT as per Table 3), GATE usually outperforms at deeper layers.

| Data | $L = 2$ | | $L = 5$ | | $L = 10$ | | $L = 20$ | |
|---|---|---|---|---|---|---|---|---|
| | GATE$_S$ | GATE | GATE$_S$ | GATE | GATE$_S$ | GATE | GATE$_S$ | GATE |
| Wisc. | **80.4** | 70.5* | **70.5** | 60.7* | **62.7** | 58.8 | **62.7** | 60.7 |
| Texas | 67.6* | **67.6*** | **67.6*** | 67.6* | 62.2* | 62.3* | 62.1* | 64.9 |
| Actor | **32.2** | 31.6 | 27.5 | **29.2** | 27.4 | **27.9** | 24.6 | **29.4** |
| Cora | **81.0*** | 80.8 | **80.8 *** | 80.4 | **80.0*** | 79.2 | 77.2* | **79** |
| Cite. | 67.6* | **68.3** | **68.7*** | 67.8 | **67.6*** | 67.6 | 67.1* | **69.2** |

## C.3 FURTHER ANALYSIS OF $\alpha$ COEFFICIENTS

We present the analysis of $\alpha$ coefficients learned for some experiments in the main paper that were deferred to the appendix due to space limitations.

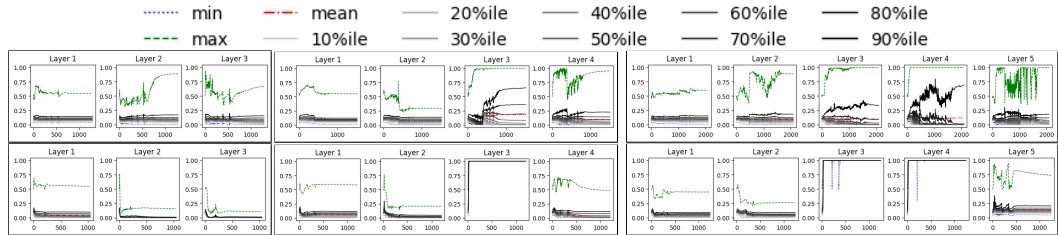

Figure 6: Distribution of $\alpha_{vv}$ against training epoch for the neighbor-dependent learning problem with $k = 3$. Rows: GAT (top) and GATE (bottom) architecture. Columns (left to right): 3, 4, and 5 layer models. While GAT is unable to switch off neighborhood aggregation in any layer, only 3 layers of the 4 and 5 layer models perform neighborhood aggregation.

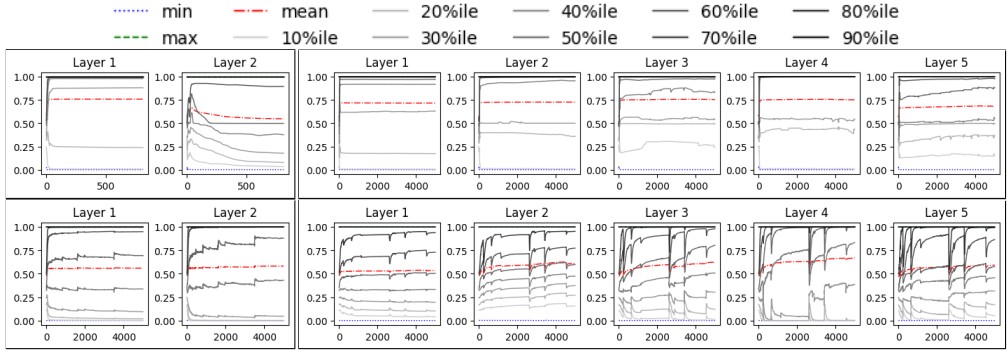

Figure 7: Distribution of $\alpha_{vv}$ against training epoch of 2-layer (left) and 5-layer (right) GAT networks for heterophilic datasets Texas (top) and Actor (bottom) 2-layer modes. Despite having connections to unrelated neighbors, GAT is unable to switch off neighborhood aggregation.

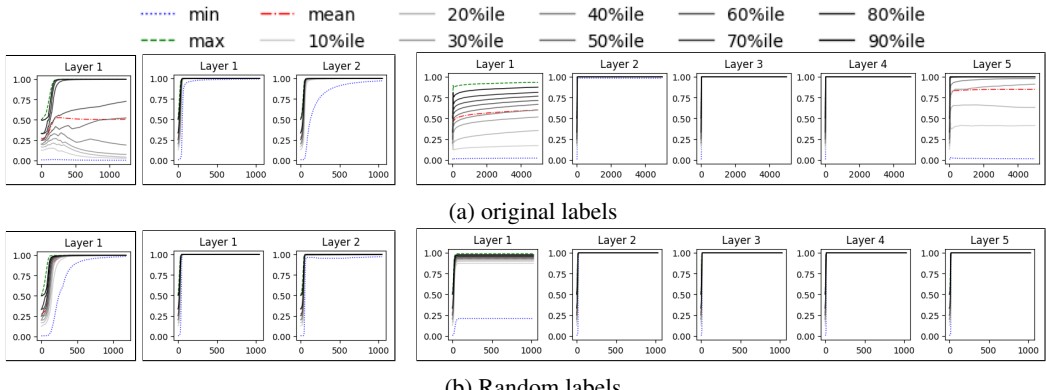

Figure 8: Distribution of $\alpha_{vv}$ against training epoch for the self-sufficient learning problem using Cora graph structure with original (top) and random (bottom) node labels and input node features as a one-hot encoding of labels. Left to right: 1, 2, and 5 layer GATE$_S$ models that all 100% test accuracy except in the case of 5 layer model using original labels. In this case, although a training accuracy if 100% is achieved at 32 epochs with test accuracy 97.3%, a maximum test accuracy of 98.4% is reached at 7257 epochs. Training the model to run to 15000 epochs only increases it to 98.4%. An increased learning rate did not improve this case. However, we also run the GAT model for 15000 epochs for this case, and it achieves 85.9% test accuracy at epoch 47 where the model achieves 100% accuracy and only achieves a maximum test accuracy of 89.3% briefly at epoch 8.

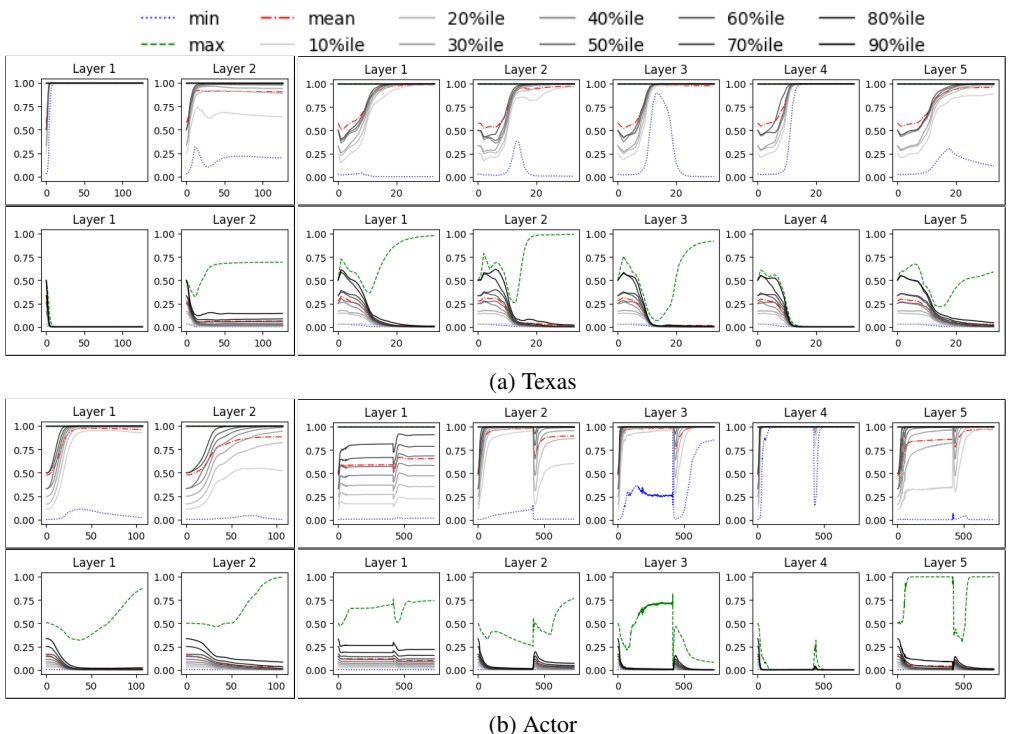

Figure 9: Distributions of $\alpha_{vv}$ (top) and $\alpha_{uv}$ (bottom) against training epoch of 2-layer (left) and 5-layer (right) GATE$_S$ networks for heterophilic datasets Texas (a) and Actor (b). The skewed distribution of $\alpha_{uv}$ shows that some level of contribution from neighbors in certain layers( which may be task-relevant) is retained.

## C.4 AGGREGATE RESULTS

For the neighbor-dependent task, we analyzed the detailed learning behavior of GAT and GATE networks in Table 2 for one run of each experiment. Here, for the same experiments, we report the average test accuracy at max. validation accuracy across 5 runs in Table 8.

Table 8: Mean test accuracy (%) $\pm 95\%$ confidence interval over 5 runs achieved for the neighbor-dependent task constituting synthetic datasets. In all cases, a GATE variant outperforms the GAT variants.

| $k$ | $L$ | $\text{GAT}_S$ | GAT | $\text{GATE}_S$ | GATE |
|---|---|---|---|---|---|
| | 1 | $93.60 \pm 1.26$ | $92.32 \pm 1.27$ | $\mathbf{96.40 \pm 0.70}$ | $93.52 \pm 1.27$ |
| 1 | 2 | $93.52 \pm 0.73$ | $92.72 \pm 2.69$ | $\mathbf{97.92 \pm 0.79}$ | $94.64 \pm 2.10$ |
| | 3 | $88.16 \pm 4.86$ | $91.76 \pm 3.35$ | $92.08 \pm 4.59$ | $\mathbf{94.00 \pm 1.54}$ |
| | 2 | $90.40 \pm 1.30$ | $87.68 \pm 1.58$ | $\mathbf{93.84 \pm 0.51}$ | $88.72 \pm 2.50$ |
| 2 | 3 | $82.16 \pm 4.45$ | $88.88 \pm 2.12$ | $85.76 \pm 2.54$ | $\mathbf{93.44 \pm 3.27}$ |
| | 4 | $84.00 \pm 5.00$ | $83.04 \pm 4.75$ | $\mathbf{89.20 \pm 2.33}$ | $87.76 \pm 2.36$ |
| | 3 | $84.32 \pm 3.18$ | $83.84 \pm 2.69$ | $87.52 \pm 1.82$ | $\mathbf{88.64 \pm 2.00}$ |
| 3 | 4 | $71.36 \pm 3.88$ | $75.92 \pm 7.63$ | $\mathbf{89.20 \pm 1.04}$ | $88.96 \pm 0.51$ |
| | 5 | $80.16 \pm 4.75$ | $83.92 \pm 2.16$ | $86.08 \pm 0.79$ | $\mathbf{87.84 \pm 1.59}$ |

## C.5 COMPARISON WITH OTHER GNNS

Other GNN architectures could potentially switch off neighborhood aggregation, as we show here. However, they are less flexible in assigning different importance to neighbors, suffer from over-smoothing, or come at the cost of an increased parameter count by increasing the size of the hidden dimensions (e.g. via a concatenation operation). We evaluate the performance of three such architectures that, in principle, employ different aggregation methods, which are likely to be capable of switching off neighborhood aggregation, on synthetic datasets empirically and discuss their ability or inability to switch off neighborhood aggregation qualitatively as follows.

$\omega$**GAT** (Eliasof et al., 2023) introduces an additional feature-wise layer parameter $\omega$ that can, in principle, switch off neighborhood aggregation by setting $\omega$ parameters to 0, in addition to the attention mechanism based on GAT. However, in practice, as we verify on our synthetic dataset in Figure 10, it is unable to effectively switch off neighborhood aggregation. Although it outperforms GAT, it is still substantially worse than GATE, especially for the deeper model due to unnecessary neighborhood aggregations. Another architecture based on graph attention, superGATKim & Oh (2021), falls under the paradigm of structural learning as it uses a self-supervised attention mechanism essentially for link prediction between nodes, and therefore its comparison with GATE is infeasible.

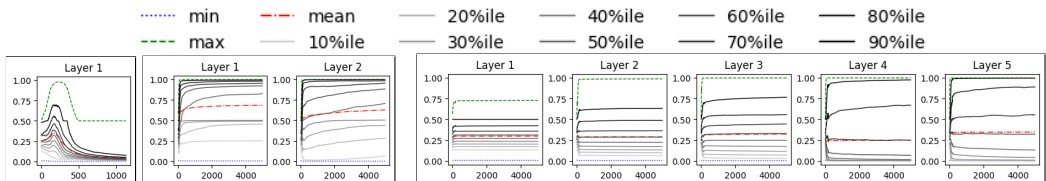

Figure 10: Distribution of $\alpha_{vv}$ against training epoch for self-sufficient learning problem using the Cora structure with random labels, where input node features are a one-hot encoding of node labels, for the $\omega$GAT architecture for the 1, 2 and 5 layer models that achieve test accuracy of $100\%$, $98.5\%$, and $49.3\%$, respectively.

**GraphSAGE** (Hamilton et al., 2018) uses the concatenation operation to combine the node's own representation with the aggregated neighborhood representation. Therefore, it is usually (but not

always) able to switch off the neighborhood aggregation for the synthetic datasets designed for the self-sufficient learning task (see Table 9). Mostly, GATE performs better on the neighbor-dependent task, in particular for deeper models, where the performance of GraphSAGE drops likely due to over-smoothing (see Table 10).

**FAGCN** (Bo et al., 2021) requires a slightly more detailed analysis. Authors of FAGCN state in the paper that: 'When $\alpha_{ij}^G \approx 0$, the contributions of neighbors will be limited, so the raw features will dominate the node representations.' where $\alpha_{ij}^G$ defined in the paper can be considered analogous to $\alpha_{ij}$ in GAT, though they are defined differently. Thus, from an expressivity point of view, FAGCN should be able to assign parameters such that all $\alpha_{ij}^G = 0$. However, we empirically observe on synthetic datasets designed for the self-sufficient learning task, values of $\alpha_{ij}^G$ do not, in fact, approach zero. Despite being unable to switch off neighborhood aggregation, FAGCN, in its default implementation, achieves $100\%$ test accuracy on the task. We discover this is so because FAGCN introduces direct skip connections of non-linearly transformed raw node features to every hidden layer. Given the simplicity of the one-hot encoded features in the datasets and the complete dependence of the label on these features, FAGCN is able to represent the desired function. In order to better judge its ability to *switch off neighborhood aggregation by setting* $\alpha_{ij}^G = 0$, we remove this skip connection. From an expressivity point of view, FAGCN should still be able to achieve $100\%$ test accuracy by using only the (non-)linear transformations of raw features initially and performing no neighborhood aggregation in the hidden layers. However, we find that FAGCN was unable to emulate this behavior in practice. For a fair comparison of the differently designed attention mechanism in FAGCN with GATE, we introduce self-loops in the data so FAGCN may also receive a node's own features in every hidden layer. Even then, FAGCN fails to achieve perfect test accuracy as shown in Table 9. Therefore, we suspect the attention mechanism in FAGCN may also be susceptible to the trainability issues we have identified for the attention mechanism in GAT. Nevertheless, the capacity of FAGCN to learn negative associations with neighboring nodes is complementary to GATE and both could be combined. It would be interesting to derive conservation laws inherent to other architectures such as FAGCN and GraphSAGE and study how they govern the behaviour of parameters. Furthermore, by design, FAGCN does not perform any non-linear transformations of aggregated neighborhood features which may be necessary in some tasks, such as our synthetic dataset for the neighbor-dependent learning task. As Table 10 shows, GATE outperforms FAGCN on such a task.

Lastly, we would like to emphasize that our aim is to provide insights into the attention mechanism of GAT and understand its limitations. While it should be able to flexibly assign importance to neighbors and the node itself without the need for concatenated representation or explicit skip connections of the raw features to every layer, it is currently unable to do so in practice. In order to verify our identification of trainability issues, we modify the GAT architecture to enable the trainability of attention parameters which control the trade-off between node features and structural information.

Table 9: Self-sufficient learning: $S, C$ and $L$ denote graph structure, number of label classes, and number of network layers, respectively. Original (Orig.) and Randomized (Rand.) labels are used for the Cora structure. The FAGCN model is implemented without skip connections from the input layer to every other layer and without any self-loops in input data, whereas FAGCN* denotes the model also without skip connections but with self-loops introduced for all nodes in input data.

| Structure | $C$ | $L$ | Max. Test Accuracy (%) | | | | |
|---|---|---|---|---|---|---|---|
| | | | GAT | GATE | FAGCN | FAGCN* | SAGE |
| Cora | O,7 | 1 | **100** | **100** | 90.1 | 97.6 | **100** |
| | | 2 | 94.6 | **100** | 94.2 | 94.9 | 98.8 |
| | | 5 | 88.5 | **99.7** | 87.1 | 89.1 | 92.4 |
| | R,7 | 1 | **100** | **100** | 61.6 | 97.8 | **100** |
| | | 2 | 57.0 | **100** | 69.2 | 70.5 | **100** |
| | | 5 | 36.7 | **100** | 21.2 | 36.7 | 99.6 |
| ER $(p = 0.01)$ | R,2 | 1 | **100** | **100** | **100** | **100** | **100** |
| | | 2 | **100** | **100** | **100** | **100** | **100** |
| | | 5 | 99.6 | **100** | 96.4 | 99.2 | **100** |
| | R,8 | 1 | 99.2 | **100** | 86.4 | 98.8 | **100** |
| | | 2 | 97.6 | **100** | 86.0 | 91.6 | **100** |
| | | 5 | 38.4 | **100** | 31.6 | 40.4 | **100** |

Table 10: Neighbor-dependent learning: $k$ and $L$ denote the number of hops aggregated in the neighborhood to generate labels, and the number of layers of the evaluated network, respectively.

| $k$ | $L$ | Max Test Accuracy (%) @ Epoch | | | |
|---|---|---|---|---|---|
| | | GAT | GATE | SAGE | FAGCN |
| 1 | 1 | 92@9564 | **93.6 @ 3511** | 93.2@2370 | 93.2@1618 |
| | 2 | 92.8@4198 | **95.6 @ 111** | 95.6@723 | 94.1@1455 |
| | 3 | 92.8@437 | **97.2 @ 82** | 96.8@100 | 81.2@573 |
| 2 | 2 | **93.2 @ 95** | 92.0@105 | 90.8@199 | 90.4@170 |
| | 3 | 93.2@856 | **95.2 @ 189** | 94.4@113 | 88.8@283 |
| | 4 | 88.4@423 | 90.4@447 | **92.4 @ 139** | 87.6@549 |
| 3 | 3 | 88.8@285 | **92.0 @ 47** | 87.6@45 | 89.2@528 |
| | 4 | 85.6@469 | **91.6 @ 139** | 88@60 | 89.2@3191 |
| | 5 | 83.6@1098 | **91.2 @ 243** | 86.0@35 | 88.8@205 |

