# OpenReview forum: "GATE: How to Keep Out Intrusive Neighbors"
_ICLR.cc/2024/Conference — Submitted to ICLR 2024_

### Official Review · Reviewer_J31R · 2023-10-31

**Soundness:** 3 good
**Presentation:** 3 good
**Contribution:** 3 good
**Rating:** 6
**Confidence:** 3

**Summary:**

The authors introduce GATE, an variant of Graph Attention Networks (GAT) which they claim alleviates the over-smoothing problem found in GNNs. The claim is that the proposed method can close the "gate" of neighborhood aggregation to irrelevant features. Experimental results on both synthetic and real world data are shown to back the claims.

**Strengths:**

The contributions of the paper are clearly mentioned.

The flow of the paper makes it easy to read.

**Weaknesses:**

A pictorial representation of the GATE mechanism would really add to the description. If it is relatively straight forward to add, please consider it. It is not absolutely necessary but may help with readers' intuition.

**Questions:**

(Table 1) Although GATE helps with the numbers for the case with more layers, typically one wouldn't use a lot of layers for datasets such as the toy one used here. And for the single layer case, GATE was not necessary to get a near 100 score.

(Table 2) Was the experiment run just once? Instead of reporting the maximum accuracy at some epoch, it would be better to look at mean accuracy over a number of runs. The latter has more information.

---

> ### Author Response · Authors · 2023-11-18
> **Response to Reviewer J31R**
>
> We thank the reviewer for the constructive feedback and insightful questions, which we answer below.
>
> - We agree with the reviewer's suggestion that a pictorial representation would help with a reader's intuition. Unfortunately, due to space constraints, it is not easy to add one to the paper.
>
>
> - As the reviewer has rightly pointed out, GATE was not necessary to get a near 100 score in the single-layer case. We highlight the importance of this observation in paragraph 6 on page 5. The key idea of the experiment is not to discover how many layers are needed for the toy case example, but rather to empirically demonstrate our theoretical insights in Section 4. Our reasoning for why GAT is unable to switch off neighborhood aggregation is related to norm constraints on the parameters imposed by an inherent conservation law in the network. For a single-layer network, the norms are not constrained and hence GAT is able to switch off neighborhood aggregation. In contrast, the 2-layer (or more) network is bound by norm constraints that prevent GAT from achieving the desired behavior. This reinforces our theoretical insight. In contrast, GATE can retain its performance for deeper models as well by switching off neighborhood aggregation in all layers.
>
>
> - All our experiments were run multiple times.  As the motivation behind designing a learning task via a synthetic dataset was to study how well GAT and GATE learn the task at hand, we decided to stick to analyzing one run of each model in detail. We compare the test accuracy of the model when it has achieved its maximum train accuracy (usually 100\% except for cases marked with asterisk) to the maximum test accuracy achieved by the model (and when). This provides an insight into whether the model is simply overfitting to the train data or really learning the task. We report the mean accuracy in Table 8 in Appendix C.4. Due to space constraints, both tables can not be added to the main text. If the reviewers suggest that the detailed analysis be moved to the appendix, it can be replaced by the aggregate results across multiple runs in the main paper.

---

> > ### Comment · Reviewer_J31R · 2023-11-18
> > **Thank you for the response.**
> >
> > I have no further questions. I have updated the score to 6.

---

### Official Review · Reviewer_8362 · 2023-11-01

**Soundness:** 3 good
**Presentation:** 3 good
**Contribution:** 3 good
**Rating:** 5
**Confidence:** 3

**Summary:**

This paper presents GATE, an extension of GAT tackling the over-smoothing issue. Speficially, the authors first show that GAT cannot effectively switching off the aggregation from irrelevant neighbors, and then modify the GAT architecture by assigning separate attention parameters for the node and the neighborhood. Experimental evaluations show the effectiveness of GATE, especially in heterophilic graphs.

**Strengths:**

- The paper is generally well-written and easy to follow.
- Theoretical analysis of the proposed method is provided.
- Extensive experiments on both synthetic and real datasets are conducted to evaluate the effectiveness of the proposed method.

**Weaknesses:**

- The theoretical analysis is restricted to the weight-sharing version. The gap between the weight-sharing version and the original version is unclear.

- Only GAT is used as the baseline in the experiments. Although the authors stated that the main focus of this paper is to show the limitations of neighborhood aggregation in GATs, it would still be meaningful to show how well does the proposed method address the over-smoothing issue in GNNs.

- Some of the figures are difficult to interpret. For example, the 20%ile - 90%ile lines in Figure 4 are diffucult to distinguish. Also, what does this figure tells?

**Questions:**

- Why did the authors report the min loss and max acc in Table 1?

---

> ### Author Response · Authors · 2023-11-18
> **Response to Reviewer 8362**
>
> We thank the reviewer for the constructive feedback and insightful questions, which we answer below.
>
> - While our derivations that lead to Insight 4.2 only consider the weight-sharing case, a similar principle also applies to GAT without weight-sharing. Experimental results in Table 1 demonstrate that GATs, both with and without weight sharing, are unable to switch off neighborhood aggregation, as they struggle to assign large magnitudes to the parameters $a$. Contrary to this, GATE, both with and without weight sharing, is able to switch off neighborhood aggregation. We have presented results on real-world datasets for both GATE and GATE$_S$ in Figure 9 and Table 6 (deferred to the appendix due to space limitation and observation of similar results) and also briefly discussed the trade-off between parameter sharing and no parameter sharing in light of training time and generalization accuracy.
>
> - We have now added a comparison of GATE with FAGCN and GraphSAGE on synthetic datasets in Appendix C.5 (Tables 9 and 10) and discuss their ability to switch off neighborhood aggregation. To the best of our knowledge, current techniques to alleviate over-smoothing in GNNs are based on regularization such as dropout and sampling, or architectural elements such as skip connections from a previous layer to the next (or from the input layer to every other layer (FAGCN), or from every layer to the last layer (GCNII)),  and concatenation of a node's own features with aggregated neighbors (GraphSAGE). These techniques are complementary and thus could be combined with GATE. Note that GATE also maintains a strong performance for deeper networks that would usually suffer from over-smoothing.
>
> - The lines from 20\%ile - 90\%ile can be distinguished by their order (bottom to top) for example in the plot for Layer 1 in the bottom right plot of Figure 4. When they are indistinguishable, for example in layers 3-4 of the same plot, it implies that they are (almost) equal (and hence do not need to be differentiated.) Section 5.2 discusses in detail the results in Figure 4 in the context of the effect of depth and how GATE offers interpretable neighborhood aggregation. However, upon the reviewer's identification of the difficulty of interpreting Figure 4 in isolation, we have added a summary of its insights from Section 5.2 to the caption of Figure 4.
>
> - Table 1 does not report the min. loss but rather the test accuracy at the epoch of min loss (most of these cases correspond to 100\% train accuracy except the entries marked with an asterisk). Because the reported results are for synthetic data designed specifically to distinguish the behavior of GAT and GATE, we believe it is important to capture the full behavior of both architectures by ensuring that even though GAT can memorize the data to achieve 100\% train accuracy, it fails to generalize, unlike GATE. The epoch information helps us to verify that despite being allowed to train for longer, GAT is unable to effectively learn the task, whereas GATE generalizes perfectly almost at the same time when it achieves 100\% training accuracy. We have added a sentence that explains this reasoning in more detail.

---

> > ### Comment · Reviewer_8362 · 2023-11-22
> > **Feedback to the authors**
> >
> > Dear authors,
> >
> > Thanks for your response. My questions are partially addressed. Given that there is sitll a gap in the theoretical analysis, I will keep my score.

---

> ### Author Response · Authors · 2023-11-22
> **Added theory for non-weight sharing case**
>
> We thank the reviewer for their response. According to their request, we have added the theoretical analysis of the non-weight-sharing case in Appendix A.1, which completes our theoretical investigation and leaves no gap.
> The flow of arguments is identical to the weight-sharing case. Yet, the analysis involves the distinction of $6$ cases, which we now discuss all in detail.
>
> We would be happy to answer any further questions upon request.

---

### Official Review · Reviewer_nehk · 2023-11-01

**Soundness:** 2 fair
**Presentation:** 4 excellent
**Contribution:** 3 good
**Rating:** 5
**Confidence:** 4

**Summary:**

The paper discusses the limitations of Graph Attention Networks (GATs) in terms of their inability to switch off unnecessary neighbors. To address this limitation, the authors propose GATE, a GAT extension that alleviates over-smoothing, enjoys the expressiveness from deeper layers, and often outperforms GATs on real-world datasets. In addition, the paper evaluates and deeply analyzes the behavior of GATE by creating synthetic datasets.

**Strengths:**

The paper is well-written and easy to follow. The proposed method is well-motivated and well-described. The GNN operator, which aggregates only relevant neighbors, is a very important component in learning graph representation. The implementation is simple but effective in real-world heterophilous benchmarks.

**Weaknesses:**

- If I understand correctly, there is a logical flaw in theoretical results. Can the authors explain the theoretical result more (conservation of norms → small ‘relative’ contributions of attention)? Why is ‘switching off neighborhood aggregation’ related to [αij/αii << 1] rather than [αij << 1]? Regardless of the self-loop, if there are large attention values in the other neighbors than j, wouldn't αij be small? The separation of attention parameters for self-loops and neighbors clearly affects [αij/αii << 1] but may not for [αij << 1].
- Most experiments are conducted on synthetic datasets and experiments on other real-world benchmarks are required (Lim, Derek, et al. and Platonov, Oleg, et al.). Synthetic datasets are useful lenses to analyze the model behavior but they are over-simplied versions of real-world datasets. However, real-world datasets used in the paper are small-scale and known to have various pitfalls (See Platonov, Oleg, et al.)
  - Lim, Derek, et al. "Large scale learning on non-homophilous graphs: New benchmarks and strong simple methods." Advances in Neural Information Processing Systems 34 (2021): 20887-20902.
  - Platonov, Oleg, et al. "A critical look at the evaluation of GNNs under heterophily: Are we really making progress?." The Eleventh International Conference on Learning Representations. 2023.
- There are existing GNNs that use different aggregators on self and neighbor nodes. Although there is no attention in model names, it is believed that they are able to perform GATE's switching off neighbors. Here are examples below. There should be qualitative and quantitative comparisons with them.
  - GraphSAGE: Hamilton, Will, Zhitao Ying, and Jure Leskovec. "Inductive representation learning on large graphs." Advances in neural information processing systems 30 (2017).
  - FAGCN: Bo, Deyu, et al. "Beyond low-frequency information in graph convolutional networks." Proceedings of the AAAI Conference on Artificial Intelligence. Vol. 35. No. 5. 2021.
  - Residual Gated Graph ConvNets: Bresson, Xavier, and Thomas Laurent. "Residual gated graph convnets." arXiv preprint arXiv:1711.07553 (2017).
- There are some missing references related to this paper’s motivation and methods:
  - Wang, Guangtao, et al. "Improving graph attention networks with large margin-based constraints." arXiv preprint arXiv:1910.11945 (2019).
  - Kim, Dongkwan, and Alice Oh. "How to Find Your Friendly Neighborhood: Graph Attention Design with Self-Supervision." International Conference on Learning Representations. 2021.
  - Fountoulakis, Kimon, et al. "Graph attention retrospective." Journal of Machine Learning Research 24.246 (2023): 1-52.
  - Lee, Soo Yong, et al. "Towards Deep Attention in Graph Neural Networks: Problems and Remedies." ICML (2023).

**Questions:**

See Weaknesses.

---

> ### Author Response · Authors · 2023-11-18
> **Response to Reviewer nehk**
>
> We thank the reviewer for the constructive feedback and insightful questions, which we answer below.
> - We would be happy to provide further explanations of the theoretical result.
>     - Firstly, note that $\alpha_{ij} << 1$ would not necessarily be sufficient to make the contribution of neighbor $j$ insignificant. For instance, nodes with a high degree $\alpha_{ij} << 1$ could hold for all $j$ and the node $i$ so that each neighbor, as well as node $i$, would still similarly impact the features of $i$. To make $j$ insignificant relative to the contribution of the node's contribution, we actually need $\alpha_{ij}/\alpha_{ii} < < 1$. Also note that the analysis of relative $\alpha_{ij}/\alpha_{ii}$ is simpler, as the normalization constant cancels out. This fact elucidates further why we need to study $\alpha_{ij}/\alpha_{ii}$ and not $\alpha_{ij}$: even if the attention parameter in another neighbor was large, this would affect the normalization constant and thus potentially also weight down $\alpha_{ii}$. Furthermore, to switch off all neighborhood aggregation, we require $\alpha_{ij}/\alpha_{ii} < < 1$ for all neighbors $j$.
>      - Next, we provide an intuition of how the limited expressiveness of attention results from the derived conservation law. Theorem 4.1 induces the corollary that $\left\lVert\mathbf{W}^{l}\right\rVert^2 = \left\lVert\mathbf{W}^{l+1}\right\rVert^2 + \left\lVert\mathbf{a}^{l}\right\rVert^2 + c$ during gradient flow.  We show that to make the contribution of all neighbors $j$ insignificant, we actually need $\left\lVert\mathbf{a}^{l}\right\rVert^2$ to be large.  The above conservation law implies that $\left\lVert\mathbf{W}^{l}\right\rVert^2$ would also need to be large and $\left\lVert\mathbf{W}^{l+1}\right\rVert^2$ relatively small. However, due to the structure of gradients in Equation (6) on page 4, this would require large `relative' gradients $\Delta \mathbf{W}^{l+1} = \nabla \mathbf{W}^{l+1}/ \mathbf{W}^{l}$ of parameters in $\mathbf{W}^{l+1}$ and small relative gradients of parameters in $\mathbf{W}^{l}$, for which the model would need to enter a less trainable regime and thus could not converge to a meaningful model.
> - We present additional results on the larger and more recent five datasets introduced in the paper `A critical look at the evaluation of GNNs under heterophily' proposed by the reviewer in Table 6 in Appendix C.1, as well as two large-scale OGB datasets: Arxiv and Product. For most of these datasets, we observe a substantial improvement of GATE over GAT.
> - We would like to emphasize that our aim is to provide insights into the attention mechanism of GATs and understand its limitations. While it should be able to flexibly assign importance to neighbors and the node itself without the need for concatenated representation or explicit skip connections of the raw features to every layer, it is currently unable to do so in practice. In order to verify our identification of trainability issues, we modify the GAT architecture to enable the trainability of attention parameters which control the trade-off between node features and structural information. Other architectures could potentially switch off neighborhood aggregation, as we show next. However, they are less flexible in assigning different importance to neighbors, suffer from over-smoothing, or come at the cost of an increased parameter count by increasing the size of the hidden dimensions (e.g. via a concatenation operation). Concretely, we have newly added a comparison of GATE with FAGCN and GraphSAGE on synthetic datasets in Appendix C.5 (Tables 9 and 10) and discuss their ability to switch off neighborhood aggregation as follows.
>      - **GraphSAGE**  uses the concatenation operation to combine the node's own representation with the aggregated neighborhood representation. Therefore, it is usually (but not always) able to switch off the neighborhood aggregation for the synthetic datasets designed for the self-sufficient learning task. Mostly, GATE performs better on the neighbor-dependent task, in particular for deeper models, where the performance of GraphSAGE drops likely due to over-smoothing.
>      - **FAGCN** requires a slightly more detailed analysis. Hence, due to space limitations, we make the complete discussion in Appendix C.5 on page 21. The key point is that FAGCN, as we observe, is unable to switch off neighborhood aggregation, particularly for deeper models.
> - The first two papers discuss training attention in GNNs with additional supervision. We have discussed the paper "How to Find Your Friendly Neighborhoods" as superGAT in Figure 10 in Appendix C.5 but have missed the citation. The last two papers suggested by the reviewer are important recent works that highlight the interest of the GNN community in understanding and addressing the limitations of graph attention. We thank the reviewer for pointing us towards further relevant literature and have added them all to our related work section.

---

> > ### Comment · Reviewer_nehk · 2023-11-21
> >
> > Dear authors. Thank you for your detailed response.
> >
> > - I have acknowledged (1) additional results on the larger and more recent five datasets (2) newly added a comparison of GATE with FAGCN and GraphSAGE (3) added relevant literature all to related work. They resolved my concerns related to each bullet.
> > - I am still reading the first bullet, and will continue the discussion about it.

---

> ### Comment · Reviewer_nehk · 2023-11-21
>
> > Firstly, note that $\alpha_{ij} << 1$ would not necessarily be sufficient to make the contribution of neighbor j insignificant. For instance, nodes with a high degree $\alpha_{ij} << 1$ could hold for all j and the node i so that each neighbor, as well as node i, would still similarly impact the features of i.
>
> I do not think it is important for an insignificant node to have a similar impact with other neighbors. What really matters is how much an insignificant node contributes to the overall neighborhoods (including self-loops). Using an example of a high-degree node i, even if insignificant j impacts similar to other neighbors, its contribution will be very small since $\alpha_{ij} << 1$. That is, the insignificant node j will have little effect on the features of node i.
>
> > Also note that the analysis of relative $\alpha_{ij}/\alpha_{ii}$ is simpler, as the normalization constant cancels out.
>
> Nevertheless, now I understand that using relative contribution $\alpha_{ij}/\alpha_{ii}$ makes an analytical interpretation simple.
>
> > even if the attention parameter in another neighbor was large, this would affect the normalization constant and thus potentially also weight down $\alpha_{ii}$. Furthermore, to switch off all neighborhood aggregation, we require $\alpha_{ij}/\alpha_{ii} < < 1$ for all neighbors j.
>
> A small $\alpha_{ii}$ can be a problem when a large portion of neighbors is insignificant. In this case, we might need to switch off all neighborhood aggregation. However, if so, we should ask a basic but important question: why do we use a neighborhood aggregation rather than just MLP on the node itself? I recommend the authors discuss this question and compare the results of MLP with GATE.

---

> > ### Author Response · Authors · 2023-11-21
> >
> > As the reviewer has rightly pointed out, an MLP should suffice when no neighborhood aggregation *at all* is required by the task. In fact, this has also been described as a motivation for this work in the introduction, where we discuss that GNNs have been shown to perform worse than MLP in some cases, due to unnecessary neighborhood aggregation. One such case is the self-sufficient learning task that we designed. Although we could expect an MLP to perform just as well on this synthetic data, in the real world, we do not know whether neighborhood aggregation (of raw features or features transformed by a perceptron or MLP), would be beneficial or not beforehand. The right task-specific architecture would need to be identified by time and resource-intensive manual tuning.
> >
> > The advantage of GATE is that it is *able to effectively learn where to place GAT or perceptron layers or a mixture, as it fits the task at hand.* Specifically, GATE emulates an MLP layer by only performing the non-linear transformation of the node's own features without then aggregating the transformed representations over the neighborhood. As we have shown, a GCN or GAT is unable to restrict the network to perform only the non-linear transformation step. Therefore, empowering the network to simulate both perceptron and standard GNN behavior allows for more expensive power instead of predefining the role. Effective perceptron and standard layers could potentially be interleaved by the model in a flexible manner, rather than being rigidly embedded in the architecture. We observe this, particularly for the heterophilic datasets in Figure 4, which we discuss in more detail in the 'interpretable neighborhood aggregation' paragraph on page 7.

---

### Official Review · Reviewer_NiY6 · 2023-11-07

**Soundness:** 2 fair
**Presentation:** 2 fair
**Contribution:** 2 fair
**Rating:** 5
**Confidence:** 3

**Summary:**

This paper proposes an extension of graph attention network called GATE to make the attention mechanism in the original GAT more powerful. The authors stand on the assumption that attention mechanisms in GAT should be able to assign heterophilic nodes low weights while homophilous nodes high weights. Through experiments and analyses, the authors discover that original GAT fails to achieve that and accordingly propose GATE, which theoretical can switching off redundant neighbors. Through experiments on datasets where raw node features contains sufficient information to infer class labels, the authors show that GATE can successfully turn off information passed from other neighbors.

**Strengths:**

1. The research problem is interesting and the proposed method is simple yet effective.

2. The proposed GATE has sufficient theoretical guarantee.

**Weaknesses:**

1. I am a little bit worrying about the motivation of this work. In the introduction, the authors claim that an effective attention mechanism should be able to block messages passed from heterophilic neighbors and constitute the remainder of this paper following this claim. However, the scenario described is only one demonstration of a successful attention mechanism. I believe that as long as the attention mechanism can project node of the same class into the similar regions on the decision manifold , it is a good attention mechanism. I think examples from this paper could be good showcases of my point [1].

2. The experiments are conducted over synthetic use cases or graphs designed for heterophily. I am wondering how's GATE's performance over benchmarks with public splits (e.g., Cora, Citeseer, and Pubmed with fixed 20 nodes per class, and arxiv and product with ogb splits).

3. I would recommend the authors to add a section at the end of section 3 to quickly summarize the key merits discussed in section 4 in case some readers don't want to read too many theoretical materials. I think in order to understand how GATE works requires a full understanding of section 4, which could be improved.

[1] Yao Ma, et al., Is Homophily a Necessity for Graph Neural Networks? ICLR'22

**Questions:**

Please refer to the weakness section.

---

> ### Author Response · Authors · 2023-11-18
> **Response to Reviewer NiY6**
>
> We thank the reviewer for the constructive feedback and insightful comments, which we address below.
>
> - The suggested paper provides an insightful discussion on 'good' and 'bad' heterophily that we have included in our literature discussion.  In similar terminology, as the reviewer has correctly pointed out, the synthetic datasets for the self-sufficient task in Section 5.1 exemplify one use case of 'bad' heterophily.  However, they were designed to verify our identification of the cause behind the limited expressivity of the attention mechanism in GATs. Our main contribution is the theoretical insight into the reasoning behind the inability of GATs to switch off neighborhood aggregation, which is rooted in norm constraints imposed by the inherent conservation law.  To verify the correctness of this insight, we have designed GATE to address the identified root cause of the problem. Similar reasoning can also be applied to nodes in the neighborhood itself (by comparing $\alpha_{uv}$ and $\alpha_{u'v}$ for $\{u',v\}\in \mathbb{N}(v)$ instead of comparing $\alpha_{uv}$ and $\alpha_{vv}$.  We would like to reiterate that our claim is that an effective attention mechanism should be able to adapt neighborhood aggregation according to the task at hand, and the presence of 'good' and 'bad' heterophily are also task-dependent.  As we show in this work, the intended capacity of the present attention mechanism to express the importance of neighboring nodes (considering a node to be a neighbor of itself by self-loops generally introduced in GATs) is impeded by trainability issues that we have verified by alleviating the trainability issue of attention parameters for one particular neighbor (the node itself). Therefore, although GATE does not eliminate the problem completely, the identification of the problem in this work serves as a stepping stone that will allow us to address the problem more generally and enable more effective attention mechanisms for GNNs in the future.
>
> - Results on Cora and Citeseer can be found in Table 3 on page 9 with fixed 20 nodes per class. In addition, we present results in Table 6 in Appendix C.1 on more datasets: OGB-arxiv, OGB-product, and larger more recent heterophilic datasets. GATE achieves a substantial improvement over GAT for most of these larger datasets, notably, for OGB-Arxiv and OGB-Product.
>
> - We thank the reviewer for their suggestion and have incorporated their feedback in the updated document.  Unfortunately, due to space limitations, we could only include a very brief two-line overview. However, we would like to encourage readers to draw their attention to Section 4, as it discusses the main contribution of our work. Detailed theoretical insights are deferred to Appendix A.

---

### Official Review · Reviewer_sCet · 2023-11-07

**Soundness:** 2 fair
**Presentation:** 3 good
**Contribution:** 2 fair
**Rating:** 6
**Confidence:** 4

**Summary:**

This paper proposes a GAT extension named GATE, which enables GATs to switch off task-irrelevant neighborhood aggregation, alleviate over-smoothing and benefit from higher depth. Experiments on real-world datasets and theoretical analyses demonstrate the model's good performance.

**Strengths:**

1. This paper is well-written and easy to follow. The organization of this paper is good.
2. The experiments are comprehensive and the theoretical analyses are solid, which makes the good performance of GATE convincing.
3. It is an interesting and novel idea to switch off task-irrelevant neighborhood aggregation for GATs.

**Weaknesses:**

1. It seems that GATE switches off task-irrelevant neighborhood by separating the parameters $a_t$ of the target nodes from $a_s$ of the source nodes. By this, the effect of target nodes is decreased by tuning $a_t$. The question is, when a neighborhood is switched off, all nodes in this neighborhood including those that are helpful to the task are synchronously switched off. Will this be harmful to the model's performance?
2. GATE is able to reduce the performance drop caused by increased depth, would it be more beneficial to introduce more hops of neighbors by using deeper GATE or it is better to use a shallower one?
3. The texts in the legends of figures are not very clear.

**Questions:**

See weaknesses.

---

> ### Author Response · Authors · 2023-11-18
> **Response to Reviewer sCet**
>
> We thank the reviewer for the constructive feedback and insightful questions, which we answer below.
>
> - While the architecture is able to completely switch off the contribution of every neighbor if that is adequate, it does not have to do this. On synthetic tasks in which neighbors do not contain any information about a node's label, the ability of GATE to switch off neighborhood aggregation improves its performance. On many real-world data sets, however, the contributions of neighbors are not synchronously switched off. We demonstrate this by comparing the distributions of $\alpha_{uv}$ where $u\in \mathbb{N}(v) \backslash v$ in Fig. 9 in Appendix C.3 for GATE that shows a skewed distribution of $\alpha_{uv}$ over all edges where not all neighbors are switched off. In principle, the self-attention mechanism and neighborhood aggregation in GATE are no different than in GAT and neighboring nodes may still be assigned different levels of importance. The advantage of GATE is that it can distinguish between node and neighbor features and weigh them more flexibly against each other. This fact explains its often superior performance to GAT, especially for heterophilic tasks.
>
> - Ideally, the architecture should be flexible enough to determine the more beneficial case among the two and this is the direction we aim to pursue with GATE as this is an interesting dilemma that manifests itself in most graph representation learning tasks and is a much broader research problem. To the best of our knowledge, so far in the literature, the number of layers of GNNs has been treated as a hyper-parameter and deeper models have been enabled by explicit skip connections and identity mappings. While these techniques are complementary, the ability of GATE to switch off neighborhood aggregation could potentially lead to the network simulating perceptron behavior in certain layers, which may then be able to learn an identity or more complex deeper non-linear transformations of nodes multiple hops away which may then again be aggregated at further depth. We discuss this briefly in the paper in Section 5.2 (paragraph: interpretable neighborhood aggregation) in light of Fig. 4. The bottom line is that this decision is task-dependent and we observe in Table 6 in Appendix C.1 that for larger graphs, deeper GATE models lead to an improvement by possibly gaining information from a larger radius in the neighborhood and/or by learning more complex features of neighbors and the node itself.
>
> - We thank the reviewer for the feedback and would be happy to improve the legends of our figures. It would be really helpful if the reviewer could be slightly more specific with an example of a figure and whether it's a readability problem or the text in the legend is ambiguous. This would allow us to make a more satisfactory improvement.

---

### Author Response · Authors · 2023-11-18
**Response Summary**

We thank the reviewers for their constructive feedback and insightful questions. Taking into account the collective feedback of all five reviewers, we summarize the updates to the paper as follows:

1. We conduct additional experiments as requested by the reviewers to demonstrate the effectiveness of GATE on seven more real-world benchmark datasets (see Table 6 in Appendix C.1).
2. We present an empirical comparison of GATE with two other GNN architectures and qualitatively analyze their potential to switch off neighborhood aggregation (see Appendix C.5 and Tables 9 and 10 therein).
3. We include further relevant literature suggested by the reviewers in our discussion of related work.
4. We extend the theoretical analysis of Insight 4.2 to the more general case of GATs without weight-sharing (see Appendix A.1).

We look forward to an engaging discussion with the reviewers and would be happy to answer any further questions.

---

### Author Response · Authors · 2023-11-22
**Request to review responses**

Dear Reviewers and Area Chair,

We kindly request that you review our (latest) responses and the revision at your earliest convenience since today (Nov 22nd) marks the end of our interaction period.

We have responded to questions and incorporated suggestions in the revised paper, as well as the requested additional theoretical analysis, experimental results, and literature discussion.

We sincerely thank you for your time and efforts in reviewing our paper, and your insightful and constructive comments.

Thank you, Authors

---

### Meta-Review · Area_Chair_fKQH · 2023-12-05

**Metareview:**

The paper introduces a new method called GATE, which is an extension of Graph Attention Networks (GAT). GATE aims to address limitations in GAT, specifically related to unnecessary neighbor handling. The proposed method aims to deactivate the aggregation of task-irrelevant neighbors, reduce over-smoothing, and take advantage of increased depth.
However, there are some concerns regarding the basic motivation behind the paper. It is recommended that the authors further discuss the necessity of GATE, including the potential problems that may arise from not deactivating the aggregation from task-irrelevant neighbors in critical applications and scenarios. Providing more examples of the problems that GATE can solve would help readers understand the significance of this work.
Furthermore, although the authors emphasize that the contribution of this paper lies in the theoretical insights into the inability of GATs to deactivate neighborhood aggregation, this aspect has not been well-received by the reviewers. Providing a more intuitive explanation of the existing theoretical analysis may help readers better understand the contribution of this work.

**Justification For Why Not Higher Score:**

The motivation is not well supported. The existing presentation tends to focus on the technical improvement of GAT in a certain scenario but lacks discussion on what key problems it can solve. This makes the overall significance relatively weak. Additionally, providing an intuitive explanation of the existing theoretical analysis may also help readers better understand the contribution of this work.

**Justification For Why Not Lower Score:**

N/A

---

### Decision · Program_Chairs · 2024-01-16

Reject